# WeTok: Powerful Discrete Tokenization for High-Fidelity Visual Reconstruction

**Shaobin Zhuang**[1,♠]    **Yiwei Guo**[3,♠]    **Fangyikang Wang**[4,♠]    **Canmiao Fu**[2]
**Zhipeng Huang**[2]    **Zeyue Tian**[5,♠]    **Xiaohui Li**[1]    **Ying Zhang**[2]    **Chen Li**[2]    **Yali Wang**[3,6,*]

[1] Shanghai Jiao Tong University    [2] WeChat Vision, Tencent Inc.
[3] Shenzhen Key Lab of Computer Vision and Pattern Recognition, Shenzhen Institutes of Advanced Technology, Chinese Academy of Sciences    [4] Zhejiang University
[5] Hong Kong University of Science and Technology    [6] Shanghai AI Laboratory

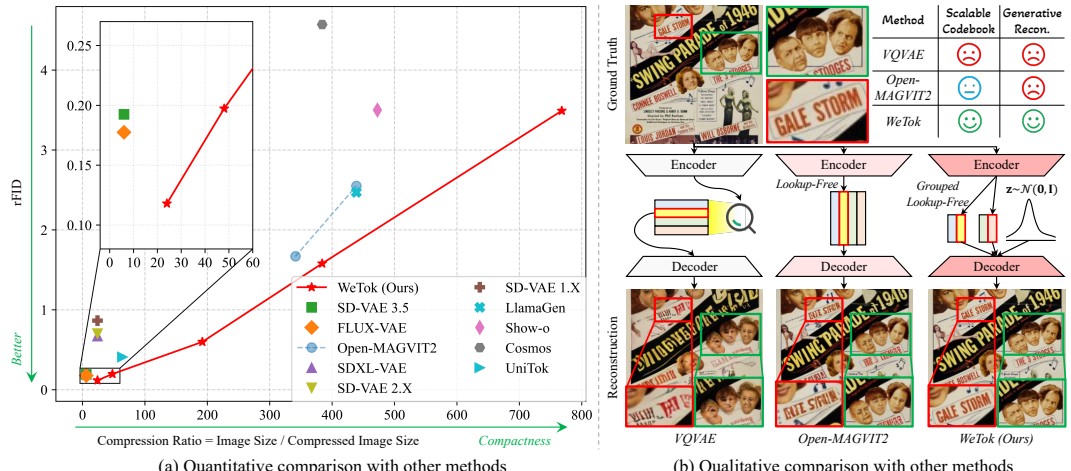

(a) Quantitative comparison with other methods

(b) Qualitative comparison with other methods

Figure 1: **Zero-shot reconstruction comparison with state-of-the-art tokenizers.** (a) Our WeTok establishes a new state-of-the-art trade-off between compression and reconstruction performance among the compared methods. (b) WeTok achieves a significant improvement in reconstruction quality over previous discrete tokenizers such as VQVAE and Open-MAGVIT2.

## Abstract

Visual tokenizer is a critical component for vision generation. However, the existing tokenizers often face unsatisfactory trade-off between compression ratios and reconstruction fidelity. To fill this gap, we introduce a powerful and concise **WeTok** tokenizer, which surpasses the previous leading tokenizers via two core innovations. (1) Group-wise lookup-free Quantization (GQ). We partition the latent features into groups, and perform lookup-free quantization for each group. As a result, GQ can efficiently overcome memory and computation limitations of prior tokenizers, while achieving a reconstruction breakthrough with more scalable codebooks. (2) Generative Decoder (GD). Different from prior tokenizers, we introduce a generative decoder with a prior of extra noise variable. In this case, GD can probabilistically model the distribution of visual data conditioned on discrete tokens, allowing WeTok to reconstruct visual details, especially at high compression ratio. On the ImageNet 50k validation set, at a high-fidelity setting, WeTok achieves a record-low zero-shot rFID of **0.12**, outperforming leading continuous tokenizers like FLUX-VAE ($0.18$) and SD-VAE 3.5 ($0.19$) with **400**% compression ratio. Furthermore, in a high-compression regime, WeTok achieves a zero-shot rFID of $3.49$ at a **768**× compression ratio, substantially surpassing Cosmos, which scores $4.57$ at only **50**% our compression ratio. Code and models are available: https://github.com/zhuangshaobin/WeTok.

---

*Corresponding author. ♠ Work done as interns at WeChat Vision, Tencent Inc.

# 1 INTRODUCTION

In visual generation, the high computational cost of pixel-based data is a central challenge (Chen et al., 2020b; Rombach et al., 2022b). Visual tokenizer is a key solution that uses an encoder to compress image into a compact latent representation and a decoder to reconstruct it (Kingma & Welling, 2013; Rezende et al., 2014), allowing generative models to operate efficiently in latent space (Rombach et al., 2022a). These tokenizers are broadly divided into two categories: *continuous* (Kingma & Welling, 2013) and *discrete* (Van Den Oord et al., 2017). Continuous tokenizers map images to a continuous latent space, while discrete tokenizers employ a quantizer to produce a finite set of codes. This architectural difference introduces a critical trade-off. Discrete tokenizers can achieve a higher compression ratio, but this efficiency often comes at the cost of lower reconstruction fidelity compared to continuous methods. This leads to a natural question: *Can we build a discrete tokenizer that can maintain high compression as well as achieve high-fidelity reconstruction?*

To achieve this goal, two critical issues must be resolved. (1) **Scalable Codebook.** To minimize quantization error of discrete tokenizers, the existing methods attempt to enlarge the codebook (Yu et al., 2024a; Zhao et al., 2024c; Sun et al., 2024). In particular, the Lookup-Free Quantization (LFQ) (Yu et al., 2024a) quantizes the latent features directly, which largely increases the codebook size for better reconstruction. However, the substantial memory and computational overhead required to manage such a large codebook during training hinders further scalability. (2) **Generative Modeling.** Discrete tokenizers are inherently deterministic. Rather than modeling data distribution of images, decoder is trained to reconstruct the expected value of images (Esser et al., 2020), corresponding to the latent codes from encoder. Such a manner is limited to capture rich diversity and fine details in the original images, leading to unsatisfactory reconstruction, particularly at high compression ratios.

To fill the gap, we introduce a powerful discrete tokenizer, **WeTok**, which consists of two concise designs to solve the issues above. First, we develop a Group-Wise Lookup-Free Quantization (GQ), which groups the latents and employs LFQ for each group. As shown in Tab. 1 and Fig. 3, GQ addresses the challenge in LFQ where entropy loss (Chang et al., 2022; Jansen et al., 2019) causes memory usage to grow with the codebook, while yielding superior reconstruction performance. Furthermore, we analyzed that the theoretical error caused by GQ is strictly smaller than that of BSQ. Second, we introduce the GAN-style generator into the decoder (GD). In Fig. 7, GD effectively models data distribution of images, allowing to reconstruct visual details at high compression ratios.

Finally, we conduct extensive experiments on mainstream benchmarks, via scaling WeTok across group size, model size, and training data size. Moreover, we pre-train our WeTok on a 400M general-domain dataset across multiple compression ratios. As illustrated in Fig. 1 (a), WeTok consistently outperforms the state-of-the-art continuous and discrete tokenizers with a **400**% compression ratio, e.g., rFID on ImageNet 50k validation set: **WeTok: 0.12 *vs.* FLUX-VAE: 0.18** (Batifol et al., 2025) *vs.* **SD-VAE 3.5: 0.19** (Esser et al., 2024a). Furthermore, our highest compression model also achieves the superior reconstruction performance, e.g., rFID on ImageNet 50k validation set: WeTok: $3.59$ *vs.* Cosmos: $4.57$ (Agarwal et al., 2025), while Cosmos ($384$) only has **50% compression ratio** of our WeTok ($768$), showing effectiveness and efficiency of WeTok.

# 2 RELATED WORK

VQVAE (Van Den Oord et al., 2017) and VQGAN (Esser et al., 2021) employ vector-quantization (VQ) to transform visual input into discrete tokens. But they both suffer from low reconstruction quality caused by instability of the codebook utilization. To overcome these drawbacks, one line of work introduces optimization strategies or modules to improve performance (Lee et al., 2022b; Shi et al., 2024; Zhu et al., 2024; Yu et al., 2024c). Another line of work focuses on scaling up the codebook size by grouping codebooks (Ma et al., 2025; Jia et al., 2025; Zhang et al., 2025; Bai et al., 2024). ImageFolder (Li et al., 2024), DualToken (Song et al., 2025) and TokenFlow (Qu et al., 2024) use multiple codebooks to assist in optimizing model understanding and generation capabilities. However, VQ-based tokenizers still introduce additional costs due to the lookup operation (Yu et al., 2021b; Lee et al., 2022b; Fang et al., 2025). MAGVIT-v2 (Yu et al., 2024a) introduces Lookup-Free Quantization (LFQ) to address extra cost and proposes the entropy loss (Chang et al., 2022; Jansen et al., 2019) to ensure the utilization of the codebook. BSQ (Zhao et al., 2024b) assumes independence between the bits of the binary code to eliminate unbearable computational overhead from entropy loss, while this assumption leads to performance degradation. In contrast, WeTok does

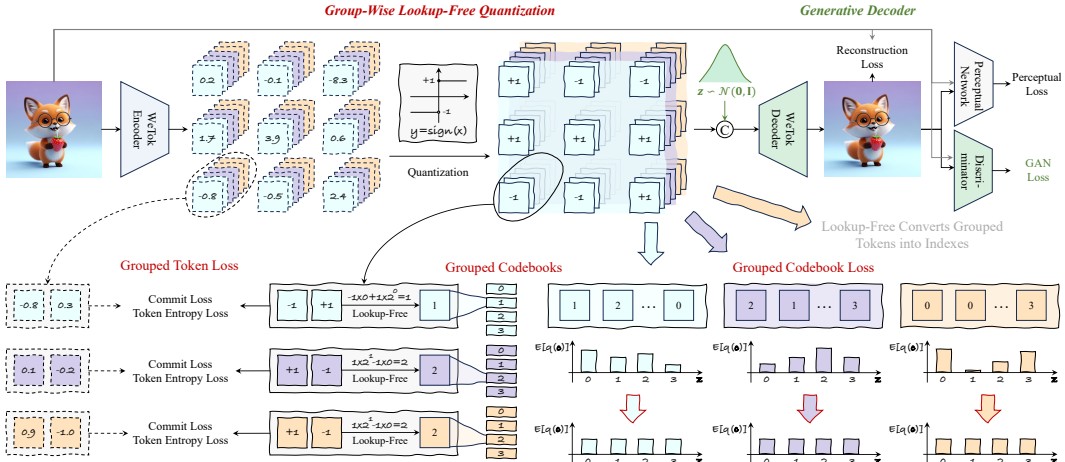

Figure 2: **WeTok with Group-Wise Lookup-Free Quantization and Generative Decoder.**

not rely on explicit codebooks, and eliminate the memory usage caused by entropy loss while having better performance than LFQ. More related works could be found in Sup. B.

DiTo (Chen et al., 2025b), consistency decoder (Betker et al.) and $\epsilon$-VAE (Zhao et al., 2024a) introduce diffusion into decoder to help visual reconstruction of continuous image tokenizer. Our WeTok is the first to introduce a generative decoder into discrete tokenizer, overcoming the instability of generative training caused by the stopping gradient estimation operation. Compare to diffusion modeling, Compared to diffusion modeling, our GAN-based WeTok achieves more efficient single-step sampling while also achieving state-of-the-art reconstruction performance.

## 3   METHOD

In this section, we first establish the necessary preliminaries for discrete tokenization. We then introduce the Group-Wise Lookup-Free Quantization (GQ) to unify Lookup-Free Quantization (LFQ) (Yu et al., 2024a) and Binary Spherical Quantization (BSQ) (Zhao et al., 2024c). Finally, we present a Generative Decoder (GD) specifically engineered for high-compression scenarios to reconstruct high-fidelity outputs from the compact representations generated by our GQ.

### 3.1   PRELIMINARIES

**Vector Quantized Variational Autoencoder (Esser et al., 2020).** VQVAE first compresses image $\mathcal{I} \in \mathbb{R}^{H \times W \times 3}$ into latent feature $\mathcal{U} = \mathcal{E}(\mathcal{I}), \mathcal{U} \in \mathbb{R}^{h \times w \times d}$, through the encoder $\mathcal{E}$. Then it is quantized into latent codes $\mathcal{Q}$ by searching the nearest neighbor in the codebook $\mathcal{C} \in \mathbb{R}^{K \times d} = [\mathbf{c}_1, \mathbf{c}_2, \ldots, \mathbf{c}_K]^\top$,

$$\mathcal{Q}[i,j] = \arg\min_{\mathbf{c} \in \mathcal{C}} \|\mathcal{U}[i,j] - \mathbf{c}\|^2, \tag{1}$$

Notice that here we define $\mathcal{Q}$ to have a backward gradient only with respect to $\mathcal{C}$, and introduce $\mathcal{U}_\mathcal{Q} = \mathcal{U} + \text{sg}[\mathcal{Q} - \mathcal{U}]$ as a variable that shares the same value as $\mathcal{Q}$ but has a backward gradient only with respect to $\mathcal{U}$. Here, $\text{sg}[\cdot]$ denotes the stop-gradient operation.

Finally, the intermediate quantized result $\mathcal{Q}$ is reconstructed into image space $\hat{\mathcal{I}} = \mathcal{G}(\mathcal{U}_\mathcal{Q})$ through the decoder $\mathcal{G}$. The loss function of VQ-VAE consists of the following five parts,

$$\mathcal{L}_{\text{VQVAE}} = \underbrace{\|\mathcal{I} - \hat{\mathcal{I}}\|^2}_{\text{Recon. Loss}} + \underbrace{\|\mathcal{Q} - \text{sg}[\mathcal{U}]\|^2}_{\text{Codebook Loss}} + \alpha \underbrace{\|\mathcal{U} - \text{sg}[\mathcal{Q}]\|^2}_{\text{Commitment Loss}} + \beta \underbrace{\mathcal{L}_{\text{LPIPS}}(\mathcal{I}, \hat{\mathcal{I}})}_{\text{Perceptual Loss}} + \gamma \underbrace{\mathcal{L}_{\text{GAN}}(\mathcal{U}_\mathcal{Q})}_{\text{GAN Loss}}, \tag{2}$$

where perceptual loss (Zhang et al., 2018) and GAN loss are introduced for better visual quality.

**Lookup-Free Quantization (Yu et al., 2024a).** LFQ introduces an implicit and learning-free codebook $\mathcal{C}_{\text{LFQ}} = \{-1, 1\}^d$ to perform lookup-free quantization on each channel of latent feature,

$$\mathcal{Q}[i,j,k] = \text{sign}(\mathcal{U}[i,j,k]). \tag{3}$$

Since the codebook in LFQ is fixed, there is no need for codebook loss during LFQ training. To address the issue of codebook utilization collapse in VQVAE, LFQ introduces the following entropy loss in replace of the commitment loss in equation 2,

$$\mathcal{L}_{\text{Entropy}}(\mathcal{U}) = \underbrace{\frac{1}{hw} \sum_{i=1}^{h} \sum_{j=1}^{w} H\left(q(\mathbf{c}|\mathcal{U}[i,j])\right)}_{\text{Token Entropy Loss}} - \zeta \underbrace{H\left(\frac{1}{hw} \sum_{i=1}^{h} \sum_{j=1}^{w} q(\mathbf{c}|\mathcal{U}[i,j])\right)}_{\text{Codebook Entropy Loss}}. \tag{4}$$

where $H(X) = -\sum_{X_i \in X} X_i \log X_i$ , $q(\mathbf{c}|\mathcal{U}[i,j])$ denote the conditional distribution of $\mathbf{c}$ given $\mathcal{U}[i,j]$ , and $\zeta$ refers to the weight of codebook entropy loss.

**Binary Spherical Quantization (Zhao et al., 2024c).** When increasing the codebook size, *i.e.*, increasing $d$, the $H(\cdot)$ calculation in the $\mathcal{L}_{\text{Entropy}}$ of LFQ leads to significant memory consumption. To alleviate this issue, BSQ processes each binary of LFQ seperately. Firstly, rewrite the token entropy loss as $\frac{1}{hw} \sum_{i=1}^{h} \sum_{j=1}^{w} \sum_{k=1}^{d} H(q_B(\mathbf{c}[k]|\mathcal{U}[i,j,k]))$. Next, the codebook entropy loss could be transformed into $\sum_{k=1}^{d} H(q_B(\mathbf{c}[k]|\frac{1}{hw} \sum_{i=1}^{h} \sum_{j=1}^{w} \mathcal{U}[i,j,k]))$ by assuming the approximation $\frac{1}{hw} \sum_{i=1}^{h} \sum_{j=1}^{w} q(\mathbf{c}|\mathcal{U}[i,j])) \approx \prod_{k=1}^{d} \frac{1}{hw} \sum_{i=1}^{h} \sum_{j=1}^{w} q_B(\mathbf{c}[k]|\mathcal{U}[i,j,k]))$. Both operations reduce the variable space of the $H(\cdot)$ calculation from $\{-1,1\}^d$ to the linear combination of $d$ $\{-1,1\}$, significantly decreasing memory consumption. However, as the approximation for the codebook entropy loss would introduce additional errors, which leads to performance degradation.

## 3.2 GROUP-WISE LOOKUP-FREE QUANTIZATION

As shown in Eq. 4, the computational cost of $H(\cdot)$ increases linearly with the codebook. To address it and the optimization error of BSQ, we group the latents and perform LFQ. As shown in Fig. 2, we group the latent features in channel dimension, reshape $\mathcal{U}$ into $\mathcal{U}_G \in \mathbb{R}^{h \times w \times g \times d'}$, where $d = gd'$ and $g$ and $d'$ represent the number and channel of groups. For $k$-th group, there is a non-learnable grouped codebook $\mathcal{C}_{\text{GQ},k} = \{-1,1\}^{d'}$. The conditional probability can be reformulated as

$$q(\mathbf{c}|\mathcal{U}[i,j]) = \prod_{k=1}^{g} q_G(\mathbf{c}_k|\mathcal{U}_G[i,j,k]), \tag{5}$$

where $\mathbf{c}_k$ refers to the $k$-th latent code after $\mathbf{c}$ is divided into $g$ parts, *i.e.*, $\mathbf{c}_k = \mathbf{c}[(k-1)d'+1 : kd']$. Considering the additivity of entropy and Eq. 5, we can rewrite the token entropy loss term as

$$\mathcal{L}_{\text{Token Entropy Loss}} = \frac{1}{hw} \sum_{i=1}^{h} \sum_{j=1}^{w} H(q(\mathbf{c}|\mathcal{U}[i,j])) = \frac{1}{hw} \sum_{i=1}^{h} \sum_{j=1}^{w} \sum_{k=1}^{g} H(q_G(\mathbf{c}_k|\mathcal{U}_G[i,j,k])). \tag{6}$$

We first transform the token entropy loss from the $\{-1,1\}^d$ space into a linear combination of $g$ $\{-1,1\}^{d'}$ spaces. This change eliminates the token entropy loss as the memory bottleneck.

For the codebook entropy loss, the $H(\sum \cdot)$ operation prevents us to decompose the $\{-1,1\}^d$ space into a linear combination of multiple subspaces. We propose the assumption that

$$\sum_{i=1}^{h} \sum_{j=1}^{w} q(\mathbf{c}|\mathcal{U}[i,j]) = \sum_{i=1}^{h} \sum_{j=1}^{w} \prod_{k=1}^{g} q_G(\mathbf{c}_k|\mathcal{U}_G[i,j,k]) \approx \prod_{k=1}^{g} \sum_{i=1}^{h} \sum_{j=1}^{w} q_G(\mathbf{c}_k|\mathcal{U}_G[i,j,k]). \tag{7}$$

Thus, we could transform the codebook entropy loss into

$$\mathcal{L}_{\text{Codebook Entropy Loss}} = H(\frac{1}{hw} \sum_{i=1}^{h} \sum_{j=1}^{w} q(\mathbf{c}|\mathcal{U}[i,j])) = \sum_{k=1}^{g} H(\frac{1}{hw} \sum_{i=1}^{h} \sum_{j=1}^{w} q_G(\mathbf{c}_k|\mathcal{U}_G[i,j,k])). \tag{8}$$

Similar to the token entropy loss, the codebook entropy loss is converted from $\{-1,1\}^d$ space into a grouped form, which eliminates the memory bottleneck caused by this term. GQ provides a tunable trade-off between approximation accuracy and memory cost by $g$. Experiments in Sec. 4.1 demonstrate that by selecting an appropriate $g$, GQ significantly reduces memory overhead while

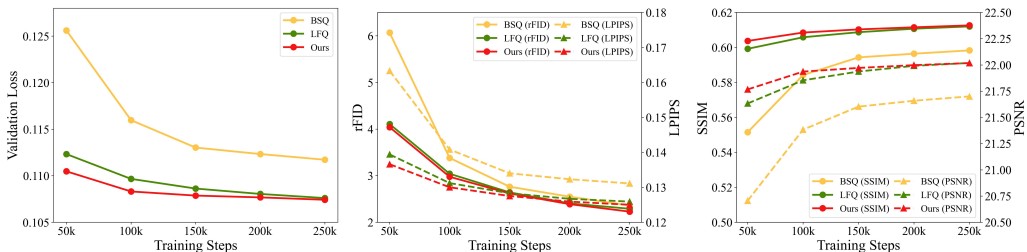

Figure 3: **Quantization method ablation.** GQ and LFQ are significantly better than BSQ.

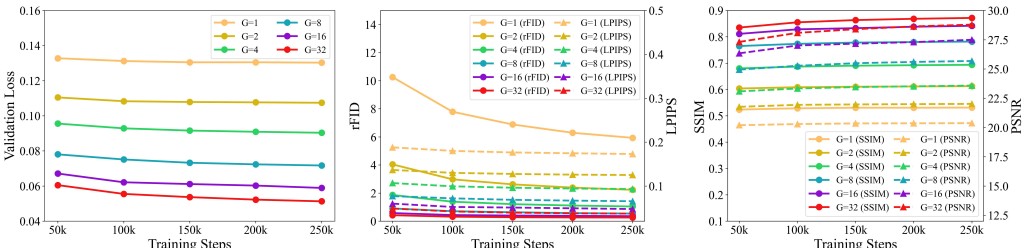

Figure 4: **Number of group ablation.** $G$ refers to the number of group. The reconstruction performance of the model increases significantly with the increase of $G$.

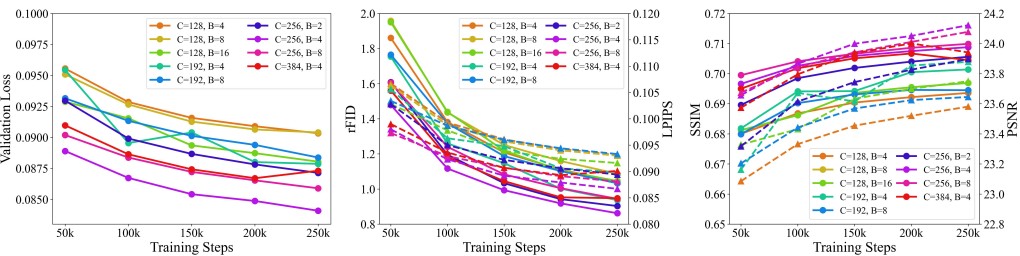

Figure 5: **Model architecture ablation.** $C$ and $B$ refer to the number of base channel and residual block respectively. $C = 256$ and $B = 4$ achieve the best reconstruction performance.

introducing minimal optimization error. This allows GQ to surpass the LFQ and BSQ. Furthermore, this tunable design provides the flexibility to scale the codebook to a virtually unlimited size.

**Proposition 3.1.** *For any choice of group $G$, the codebook entropy approximation error (as in Eq. 7) of our GQ method is smaller than that of the BSQ method.*

**Remark 3.2.** *The detailed proof and definition of the approximation error of the Proposition 3.1 is deferred to Sup. A. We mainly adopt the order theory in abstract algebra to derive such a conclusion.*

### 3.3 GENERATIVE DECODER

Unlike the continuous tokenizer, the decoders in previous discrete (Esser et al., 2021; Yu et al., 2024a) tokenizers fit deterministic transformations. Although GAN loss is employed during training as shown in Eq. 2 and its specific form is as follows

$$\mathcal{L}_{\text{GAN}}(\mathcal{U}_{\mathcal{Q}}) = \log(1 - \mathcal{D}(\mathcal{G}(\mathcal{U}_{\mathcal{Q}}))), \tag{9}$$

where $\mathcal{D}$ is the discriminator. However, the GAN loss only serves to assist in improving the perceptual quality. In high compression ratio scenarios, the correspondence between $\mathcal{U}_{\mathcal{Q}}$ and the ground truth is likely not unique. In such cases, what we need is a generative decoder. As shown in Fig. 2, we randomly sample $\mathbf{z} \in \mathcal{N}(\mathbf{0}, \mathbf{I})$, concatenate it with $\mathcal{U}_{\mathcal{Q}}$ along the channel dimension, and then feed the result into the decoder. In this way, the GAN loss is subsequently reformulated as

$$\mathcal{L}_{\text{GAN}}(\mathcal{U}_{\mathcal{Q}}) = \mathbb{E}_{\mathbf{z} \in \mathcal{N}(\mathbf{0}, \mathbf{I})}[\log(1 - \mathcal{D}(\mathcal{G}(\mathbf{z}, \mathcal{U}_{\mathcal{Q}})))], \tag{10}$$

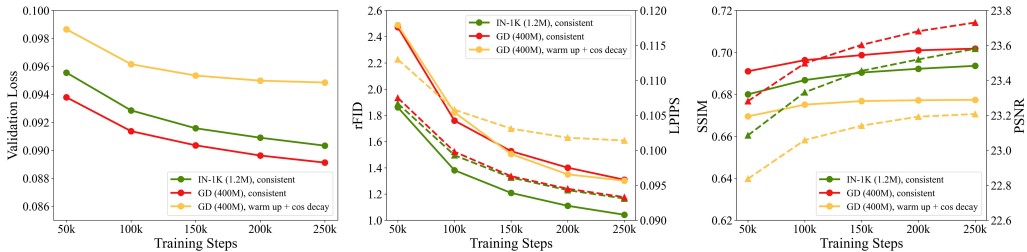

Figure 6: **Training data and learning rate schedule ablation.** GD refers to general-domain data. The model trained on general-domain data is not as good as the model trained on in-distribution data in terms of distribution fitting metrics, but has better generalization on PSNR and SSIM. The effect of consistent learning schedule is significant compared with warm up + cosine decay.

Table 1: **Memory usage ablation.** We set $d'=8$ in GQ. *OOM* refers to *out of memory*.

| Method | $d = 8$ | $d = 16$ | $d = 24$ | $d = 32$ | $d = 40$ |
|---|---|---|---|---|---|
| LFQ | 10.5 GB | 10.6 GB | *OOM* | *OOM* | *OOM* |
| BSQ | 10.5 GB | 10.5 GB | 10.6 GB | 10.6 GB | 10.6 GB |
| GQ | 10.5 GB | 10.6 GB | 10.6 GB | 10.6 GB | 10.6 GB |

Table 2: **Generative decoder modeling ablation.** *Stage2* refers to generative decoder modeling.

| Stage1 | Stage2 | rFID ↓ | LPIPS ↓ | SSIM ↑ | PSNR ↑ |
|---|---|---|---|---|---|
| ✓ | ✗ | 5.37 | 0.17 | 0.54 | 20.53 |
| ✓ | ✓ | **3.90** | **0.16** | **0.55** | **20.72** |

where $\mathcal{U}_\mathcal{Q}$ serves as a condition for the generation process. The development from Eq. 9 to Eq. 10 is not merely concatenating **z** to the input of the decoder. More importantly, the decoder shifts to modeling the transformation from Gaussian noise conditioned on $\mathcal{U}_\mathcal{Q}$ to the ground truth distribution. In previous discrete tokenizer, the decoder is trained to minimize a reconstruction loss like L2 or LPIPS. When a single, highly compressed discrete token $\mathcal{U}_\mathcal{Q}$ could correspond to multiple ground-truth images (e.g., different textures of fur , different patterns of leaves), the decoder learns to output the conditional expectation or the "average" of all these possibilities. Our GD, by incorporating **z** as an additional input, transforms the decoder into a conditional generative model. It no longer learns a one-to-one mapping but instead models the conditional distribution. The noise vector **z** allows the decoder to sample one specific, plausible instance from this distribution. This single sample can contain coherent, high-frequency details (e.g., a specific fur texture), making the output appear much more realistic and less blurry than the "average" image.

To ensure training stability, we employ a two-stage training strategy. In first stage, we train our WeTok with the reconstruction loss, *i.e.*, Eq. 2, 6 and 8. In the second stage, we adapt the model for generative tasks. Specifically, we expand the channel dimension of the `conv_in` layer in decoder and employ zero-initialization to new channel to accept **z** as additional input. This strategy ensures that at the beginning of the second stage, the decoder's behavior is identical to its pre-trained state.

## 4 EXPERIMENTS

**Datasets.** We perform large-scale training on two datasets: **(i) 1.2M** ImageNet (Russakovsky et al., 2014) training set; **(ii) 400M** general-domain dataset. For comparison, we evaluate WeTok performance on ImageNet 50k validation set and MS-COCO 2017 validation set (Lin et al., 2014). Unless otherwise stated, we conduct a series of ablation studies on the ImageNet training set. Besides, we train the class-to-image model on the ImageNet training set and test it on the validation set.

**Settings.** WeTok adopts the architecture proposed in Open-MAGVIT2 (Luo et al., 2024), employing the CNN architecture for encoder, decoder, and discriminator. Images are randomly cropped to $256 \times 256$ for training. For ablation study, all models are trained for 250K steps with Adam (Kingma & Ba, 2014) and a consistent set of hyperparameters. For large-scale training, hyperparameters are individually tuned for each model to achieve optimal performance. For class-to-image generation, we adopt the transformer architecture in LlamaGen (Sun et al., 2024). More details in Sup. E.

### 4.1 ABLATION STUDY

We conducted a comprehensive ablation study to validate the key components of WeTok. We first verify the effectiveness of our proposed GQ and GD. Then we ablate the performance improvements

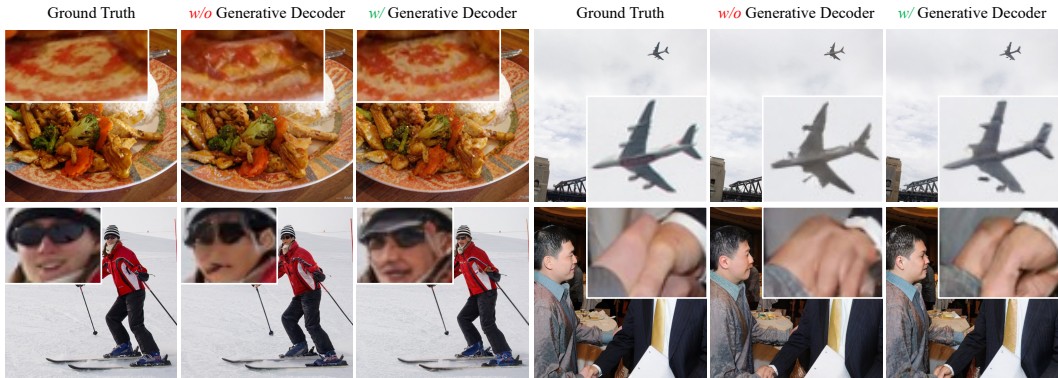

Figure 7: **Qualitative ablation of GD on MS-COCO val2017.** The images reconstructed by the model with GD are obviously more fidelity and natural.

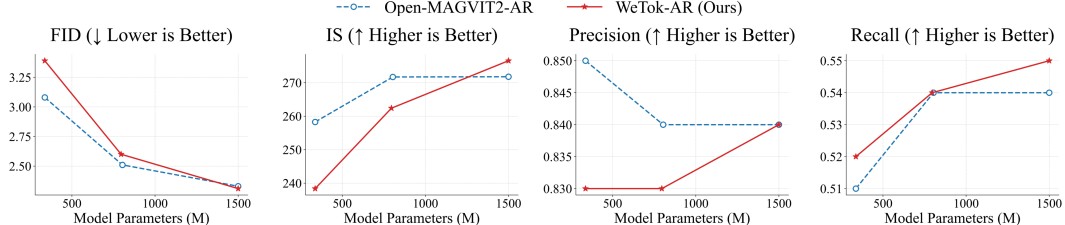

Figure 8: **Parameter ablation experiments of the Autoregressive model with WeTok.** The performance of the AR model with WeTok improves more significantly with the increase of parameters.

in various dimensions, including the number of groups, model architecture, training data, and learning rate schedule. In addition, we conduct parameter ablation experiments on the AR model based on WeTok. More details in Sup. E.1. We employ validation loss, rFID (Heusel et al., 2017), LPIPS (Zhang et al., 2018), SSIM (Wang et al., 2004), and PSNR to evaluate the quality of ablation study.

**Quantization method.** We ablate quantization methods under the same compression ratio, *i.e.*, $d=gd'=16$. As shown in Tab. 1, GQ does not introduce additional memory usage as BSQ. In Fig. 3, we set the same compression ratio for 3 different quantization methods (LFQ: $g=1$, $d'=16$; BSQ: $g=16$, $d'=1$; GQ: $g=2$, $d'=8$), GQ performs better than LFQ and far exceeds BSQ.

**Generative decoder.** When the performance of the model is saturated after the first stage of reconstruction training, we conduct the second stage of generative training. As shown in Tab. 2, the results show that GD continues to improve the reconstruction performance of the model, especially in rFID. We present qualitative ablation results in Fig. 7. After converting the decoder to a generative model, the reconstructed images are more realistic, demonstrating the effectiveness of our GD.

**Number of groups in GQ.** We increase the number of groups $g$ by a power of 2. As shown in Fig. 4, the results show that as $g$ increases, the reconstruction performance of the model continues to increase significantly and does not encounter the memory bottleneck like LFQ.

**Model architecture.** We ablate the number of base channels and residual blocks in the encoder and decoder to scale up WeTok. As shown in Fig. 5, across 9 different settings, a configuration with 256 base channels and 4 residual blocks achieves the best reconstruction performance. The encoder and decoder for this optimal architecture contain 198M and 261M parameters, respectively.

**Training data.** As shown in Fig. 6, we ablated models trained on ImageNet versus a large 400M general-domain dataset. The model trained on the general-domain data achieves a lower validation loss and higher SSIM and PSNR, but performs worse on rFID and LPIPS. We attribute to the distribution gap between the general-domain data and the in-distribution ImageNet dataset. This result highlights a trade-off between generalization and performance on in-distribution evaluation metrics.

**Learning rate schedule.** The warm-up and cosine decay learning rate schedule is widely adopted in training. However, this convention may be suboptimal for the training of discrete tokenizers. In Fig. 6, the model trained with a constant learning rate demonstrates better performance. Based on this, we adopt the constant learning rate schedule for all subsequent large-scale training in WeTok.

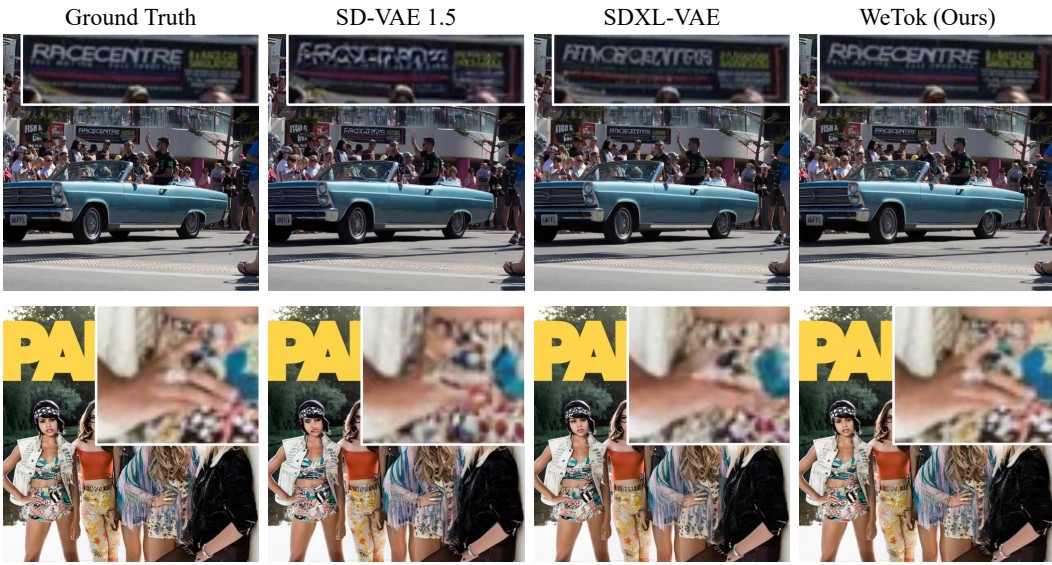

Ground Truth    SD-VAE 1.5    SDXL-VAE    WeTok (Ours)

Figure 9: **Qualitative comparison of $512 \times 512$ image reconstruction on TokBench.**

Table 3: **Reconstruction evaluation on $256 \times 256$ ImageNet 50K validation set.** All models are trained on ImageNet. WeTok achieves SOTA results on different downsampling rates. * specifies that it is obtained through our testing.

| Method | Token Type | Tokens | Ratio | Train Resolution | Codebook Size | rFID↓ | PSNR↑ | Codebook Usage↑ |
|---|---|---|---|---|---|---|---|---|
| VQGAN (Esser et al., 2020) | 2D | $16 \times 16$ | 16 | $256 \times 256$ | 1024 | 8.30 | 19.51 | – |
| VQGAN (Esser et al., 2020) | 2D | $16 \times 16$ | 16 | $256 \times 256$ | 16384 | 4.99 | 20.00 | – |
| SD-VQGAN (Rombach et al., 2022b) | 2D | $16 \times 16$ | 16 | $256 \times 256$ | 16384 | 5.15 | – | – |
| MaskGIT (Chang et al., 2022) | 2D | $16 \times 16$ | 16 | $256 \times 256$ | 1024 | 2.28 | – | – |
| ReVQ (Zhang et al., 2025) | 2D | $16 \times 16$ | 16 | $256 \times 256$ | 65536 | 2.57 | 21.69 | – |
| LlamaGen (Sun et al., 2024) | 2D | $16 \times 16$ | 16 | $256 \times 256$ | 32768 | 2.26 | 20.59 | 85% |
| LlamaGen (Sun et al., 2024) | 2D | $16 \times 16$ | 16 | $256 \times 256$ | 16384 | 2.19 | 20.79 | 97% |
| ReVQ (Zhang et al., 2025) | 2D | $16 \times 16$ | 16 | $256 \times 256$ | $2^{18}$ | 2.05 | 21.96 | – |
| SweetTok (Yu et al., 2024b) | 1D | 256 | 16 | $256 \times 256$ | 10481 | 0.73 | – | – |
| TiTok* (Yu et al., 2024b) | 1D | 256 | 16 | $256 \times 256$ | 4096 | 1.66 | 20.01 | 100% |
| FlexTok (Bachmann et al., 2025) | 1D | 256 | – | $256 \times 256$ | 64000 | 1.45 | 18.53 | – |
| VAR (Tian et al., 2024) | 2D | $16 \times 16$ | 16 | $256 \times 256$ | 4096 | – | 21.30 | 97% |
| IBQ (Shi et al., 2025) | 2D | $16 \times 16$ | 16 | $256 \times 256$ | 16384 | 1.37 | 22.35 | 96% |
| Open-MAGVIT2 (Luo et al., 2024) | 2D | $16 \times 16$ | 16 | $256 \times 256$ | $2^{18}$ | 1.17 | 22.64 | 100% |
| IBQ (Shi et al., 2025) | 2D | $16 \times 16$ | 16 | $256 \times 256$ | $2^{18}$ | 1.00 | 20.30 | 84% |
| FlowMo-Lo (Sargent et al., 2025) | 1D | 256 | – | $256 \times 256$ | $2^{18}$ | 0.95 | 22.07 | – |
| VFMTok (Zheng et al., 2025) | 1D | 256 | – | $256 \times 256$ | 16384 | 0.89 | – | 100% |
| GigaTok (Xiong et al., 2025) | 1D | 256 | – | $256 \times 256$ | 16384 | 0.79 | 21.65 | – |
| AliTok (Wu et al., 2025c) | 1D | 273 | – | $256 \times 256$ | 4096 | 0.84 | – | – |
| MGVQ (Jia et al., 2025) | 2D | $16 \times 16$ | 16 | $256 \times 256$ | $2^{52}$ | 0.64 | 23.71 | 100% |
| **WeTok (Ours)** | 2D | $16 \times 16$ | 16 | $256 \times 256$ | $2^{32}$ | **0.61** | **24.50** | **100%** |
| ViT-VQGAN (Yu et al., 2021a) | 2D | $32 \times 32$ | 8 | $256 \times 256$ | 8192 | 1.28 | – | – |
| OmiTokenizer-VQ (Wang et al., 2024a) | 2D | $32 \times 32$ | 8 | $256 \times 256$ | 8192 | 1.11 | – | – |
| LlamaGen (Sun et al., 2024) | 2D | $32 \times 32$ | 8 | $256 \times 256$ | 16384 | 0.59 | 24.45 | – |
| Open-MAGVIT2 (Luo et al., 2024) | 2D | $32 \times 32$ | 8 | $128 \times 128$ | $2^{18}$ | 0.34 | 27.02 | 100% |
| BSQ (Zhao et al., 2024b) | 1D | 1024 | – | $256 \times 256$ | $2^{18}$ | 1.14 | 25.36 | 100% |
| FlowMo-Hi (Sargent et al., 2025) | 1D | 1024 | – | $256 \times 256$ | $2^{18}$ | 0.56 | 24.93 | – |
| Selftok (Wang et al., 2025) | 1D | 1024 | – | $256 \times 256$ | $2^{15}$ | 0.54 | 26.30 | – |
| BSQ (Zhao et al., 2024b) | 1D | 1024 | – | $256 \times 256$ | $2^{36}$ | 0.45 | 28.14 | 100% |
| SweetTok (Yu et al., 2024b) | 1D | 1024 | 16 | $256 \times 256$ | 10481 | 0.37 | – | – |
| MGVQ (Jia et al., 2025) | 2D | $32 \times 32$ | 8 | $256 \times 256$ | $2^{52}$ | 0.31 | 28.42 | 100% |
| **WeTok (Ours)** | 2D | $32 \times 32$ | 8 | $256 \times 256$ | $2^{32}$ | **0.19** | **29.69** | **100%** |

**Parameter of autoregressive model.** As shown in Fig. 8, we ablate the parameter size of the WeTok-based AR model and compared it with Open-MAGVIT2-based AR models. The results show that the performance of the WeTok-based AR model is slightly inferior than the Open-

Table 4: **Zero-shot reconstruction comparison on ImageNet and MS-COCO val2017 validation set.** Our WeTok achieves the best performance on both resolution settings.

| Method | Tokenizer Type | Training Data | Ratio | Compression Ratio↑ | MS-COCO 2017 | | | Imagenet-1k | | |
|---|---|---|---|---|---|---|---|---|---|---|
| | | | | | rFID↓ | PSNR↑ | SSIM↑ | rFID↓ | PSNR↑ | SSIM↑ |
| *Resize* 256 × 256 | | | | | | | | | | |
| **WeTok (Ours)** | Discrete | 400M | 32 | **768** | **8.94** | **20.31** | **0.55** | **3.49** | **20.77** | **0.55** |
| Cosmos (Agarwal et al., 2025) | Discrete | - | 16 | 384 | 11.97 | 19.22 | 0.48 | 4.57 | 19.93 | 0.49 |
| Show-o (Xie et al., 2024) | Discrete | 35M | 16 | 473 | 9.26 | 20.90 | 0.59 | 3.50 | 21.34 | 0.59 |
| Open-MAGVIT2-I-PT (Luo et al., 2024) | Discrete | 100M | 16 | 439 | 7.93 | **22.21** | 0.62 | 2.55 | 22.21 | 0.62 |
| LlamaGen (Sun et al., 2024) | Discrete | 70M | 16 | 439 | 8.40 | 20.28 | 0.55 | 2.47 | 20.65 | 0.54 |
| **WeTok (Ours)** | Discrete | 400M | 16 | **384** | **6.55** | 21.99 | **0.63** | **1.58** | **22.38** | **0.62** |
| DALL-E dVAE (Ramesh et al., 2021a) | Discrete | 103M | 18 | 118 | 48.60 | **26.97** | 0.08 | 32.63 | **27.31** | **0.79** |
| BSQ (Zhao et al., 2024c) | Discrete | 1B | - | 219 | - | - | - | 3.81 | 24.12 | 0.66 |
| QLIP-B (Zhao et al., 2025) | Discrete | 1B | - | 219 | - | - | - | 3.21 | 23.16 | 0.63 |
| QLIP-L (Zhao et al., 2025) | Discrete | 1B | - | 168 | - | - | - | 1.46 | 25.36 | 0.69 |
| SD-VAE 1.x (Rombach et al., 2022a) | Discrete | 1B | 8 | 110 | 5.75 | 24.17 | 0.70 | 1.13 | 24.48 | 0.69 |
| **WeTok (Ours)** | Discrete | 400M | 16 | **192** | **4.41** | 24.44 | **0.74** | **0.60** | 24.77 | 0.73 |
| SD-VAE 1.x (Rombach et al., 2022a) | Continuous | 1B | 8 | 24 | 5.94 | 23.21 | 0.69 | 1.22 | 23.54 | 0.68 |
| QLIP-B (Zhao et al., 2025) | Discrete | 1B | - | 55 | - | - | - | 0.70 | 26.79 | 0.79 |
| SD-VAE 2.x (Rombach et al., 2022a) | Continuous | 6B | 8 | 24 | 4.26 | 26.62 | 0.77 | 0.70 | 26.90 | 0.76 |
| SDXL-VAE (Podell et al., 2023) | Continuous | >6B | 8 | 24 | 3.93 | 27.08 | 0.80 | 0.67 | 27.37 | 0.78 |
| UniTok (Ma et al., 2025) | Discrete | 1B | 16 | 64 | - | - | - | 0.41 | - | - |
| $\epsilon$-VAE-SD (Zhao et al., 2024a) | Continuous | - | 8 | 24 | 3.65 | 26.01 | 0.86 | 0.38 | 29.49 | 0.85 |
| **WeTok (Ours)** | Discrete | 400M | 8 | **48** | **2.18** | **29.49** | **0.89** | **0.20** | **29.63** | **0.88** |
| SD-VAE 3.5 (Esser et al., 2024b) | Continuous | - | 8 | 6 | 1.66 | 31.08 | 0.90 | 0.19 | 31.19 | 0.90 |
| FLUX-VAE (Labs, 2024) | Continuous | - | 8 | 6 | **1.35** | **32.32** | 0.93 | 0.18 | **32.74** | 0.92 |
| **WeTok (Ours)** | Discrete | 400M | 8 | **24** | 1.43 | 32.00 | **0.93** | **0.12** | 32.06 | **0.93** |
| *Original Resolution* | | | | | | | | | | |
| **WeTok (Ours)** | Discrete | 400M | 32 | **768** | **8.94** | **20.31** | **0.55** | **3.49** | **20.77** | **0.55** |
| Cosmos (Agarwal et al., 2025) | Discrete | - | 16 | 384 | 7.23 | 20.45 | 0.53 | 2.52 | 20.49 | 0.52 |
| Open-MAGVIT2-I-PT (Luo et al., 2024) | Discrete | 100M | 16 | 439 | 6.65 | 21.61 | 0.57 | 1.39 | 21.74 | 0.56 |
| **WeTok (Ours)** | Discrete | 400M | 16 | **384** | **5.30** | **21.94** | **0.59** | **0.81** | **21.99** | **0.58** |
| DALL-E dVAE (Ramesh et al., 2021a) | Discrete | 103M | 18 | 118 | 55.07 | **25.15** | **0.75** | 36.84 | 25.46 | 0.74 |
| SD-VAE 1.x (Rombach et al., 2022a) | Discrete | 1B | 8 | 110 | 6.07 | 22.54 | 0.65 | 1.23 | 22.82 | 0.64 |
| **WeTok (Ours)** | Discrete | 400M | 16 | **192** | **3.80** | 23.70 | 0.67 | **0.40** | 23.75 | 0.67 |
| SD-VAE 1.x (Rombach et al., 2022a) | Continuous | 1B | 8 | 24 | 5.94 | 21.68 | 0.64 | 1.35 | 21.99 | 0.63 |
| SD-VAE 2.x (Rombach et al., 2022a) | Continuous | 6B | 8 | 24 | 4.63 | 24.82 | 0.72 | 0.78 | 25.08 | 0.71 |
| SDXL-VAE (Podell et al., 2023) | Continuous | >6B | 8 | 24 | 4.23 | 25.11 | 0.74 | 0.72 | 25.38 | 0.73 |
| **WeTok (Ours)** | Discrete | 400M | 8 | **48** | **2.09** | **27.50** | **0.82** | **0.18** | **27.54** | **0.82** |
| SD-VAE 3.5 (Esser et al., 2024b) | Continuous | - | 8 | 6 | 1.64 | 28.35 | 0.86 | 0.24 | 28.39 | 0.86 |
| **WeTok (Ours)** | Discrete | 400M | 8 | **24** | **1.46** | **29.47** | **0.88** | **0.12** | **29.51** | **0.88** |

MAGVIT2-based AR model when the parameter size is small. However, as the parameter size increases, WeTok-based AR model surpasses the Open-MAGVIT2-based AR model in all metrics.

## 4.2 COMPARISON WITH STATE-OF-THE-ART

**Visual Reconstruction.** We first evaluate WeTok's performance in in-distribution setting on ImageNet. As shown in Tab. 3, WeTok outperforms existing methods across different downsampling ratios. Subsequently, we evaluate WeTok's performance in general-domain setting. As shown in Tab. 4, WeTok shows the state-of-the-art reconstruction performance in a wide range of compression ratio scenarios. It not only shows the strongest performance among discrete tokenizers, but also even surpasses the current strongest continuous tokenizers, FLUX-VAE (Labs, 2024) and SD-VAE 3.5 (Esser et al., 2024a). As shown in Fig. 9 and 14, we present qualitative comparison results on the TokBench (Wu et al., 2025b), our WeTok outperforms the widely used SDXL-VAE (Podell et al., 2023) and SD-VAE 1.5 (Rombach et al., 2022b) under the same compression ratio.

**Iterative Visual Reconstruction.** The use of a tokenizer for iterative image compression-decompression is a promising approach for information transmission (Joshi et al., 2000). As shown in Fig. 10, we compare WeTok (192 compression ratio) with SOTA tokenizers (Labs, 2024; Esser et al., 2024b) on iterative image compression-decompression. We surprisingly find that while reconstructions from leading models like FLUX-VAE and SD-VAE 3.5 collapse after iterations, WeTok's outputs are remarkably robust and converge to a fixed value. More details in Sup. C.

**Visual Generation.** To evaluate WeTok's capabilities for visual generation, we modifys LlamaGen like Open-MAGVIT2. We employ the in-distribution WeTok with $16\times$ downsampling in Tab. 3 as

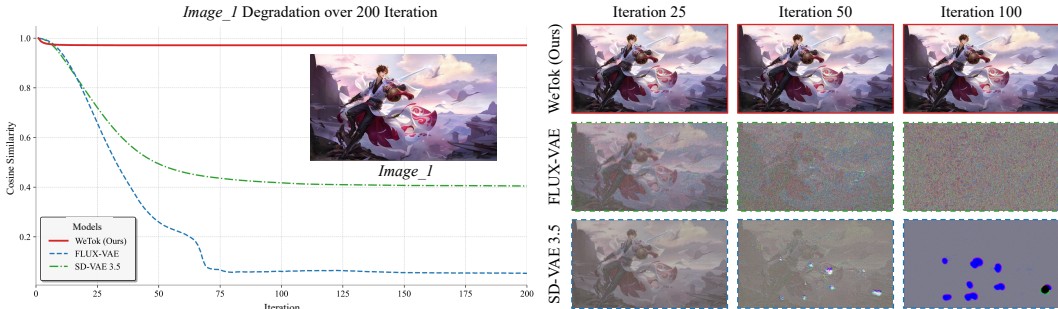

Figure 10: **Qualitative and quantitative comparison of iterative image reconstruction.**

Table 5: **Class-conditional generation on $256 \times 256$ ImageNet.** ∗ specifies the generated images are $384 \times 384$ and are resized to 256×256 for evaluation.

| Type | Model | #Para. | FID↓ | IS↑ | Precision↑ | Recall↑ |
|---|---|---|---|---|---|---|
| Diffusion | ADM (Dhariwal & Nichol, 2021) | 554M | 10.94 | 101.0 | 0.69 | 0.63 |
| | CDM (Ho et al., 2021) | — | 4.88 | 158.7 | — | — |
| | LDM-4 (Rombach et al., 2022b) | 400M | 3.60 | 247.7 | — | — |
| | DiT-XL/2 (Peebles & Xie, 2022) | 675M | 2.27 | 278.2 | 0.83 | 0.57 |
| | SiT-XL/2 (Ma et al., 2024) | 675M | 2.06 | 277.50 | 0.83 | 0.59 |
| AR | VQGAN (Esser et al., 2020) | 227M | 18.65 | 80.4 | 0.78 | 0.26 |
| | VQGAN (Esser et al., 2020) | 1.4B | 15.78 | 74.3 | — | — |
| | VQGAN-re (Esser et al., 2020) | 1.4B | 5.20 | 280.3 | — | — |
| | ViT-VQGAN (Yu et al., 2021a) | 1.7B | 4.17 | 175.1 | — | — |
| | ViT-VQGAN-re (Yu et al., 2021a) | 1.7B | 3.04 | 227.4 | — | — |
| | RQTran. (Lee et al., 2022a) | 3.8B | 7.55 | 134.0 | — | — |
| | RQTran.-re (Lee et al., 2022a) | 3.8B | 3.80 | 323.7 | — | — |
| | LlamaGen-L∗ (Sun et al., 2024) | 343M | 3.07 | 256.06 | 0.83 | 0.52 |
| | LlamaGen-XL∗ (Sun et al., 2024) | 775M | 2.62 | 244.08 | 0.80 | 0.57 |
| | LlamaGen-XXL∗ (Sun et al., 2024) | 1.4B | 2.34 | 253.90 | 0.80 | 0.59 |
| | LlamaGen-L (Sun et al., 2024) | 343M | 3.80 | 248.28 | 0.83 | 0.51 |
| | LlamaGen-XL (Sun et al., 2024) | 775M | 3.39 | 227.08 | 0.81 | 0.54 |
| | LlamaGen-XXL (Sun et al., 2024) | 1.4B | 3.09 | 253.61 | 0.83 | 0.53 |
| | UniTok (Ma et al., 2025) | 1.4B | 2.51 | 216.7 | 0.82 | 0.57 |
| | Open-MAGVIT2-AR-B (Luo et al., 2024) | 343M | 3.08 | 258.26 | 0.85 | 0.51 |
| | Open-MAGVIT2-AR-L (Luo et al., 2024) | 804M | 2.51 | 271.70 | 0.84 | 0.54 |
| | Open-MAGVIT2-AR-XL (Luo et al., 2024) | 1.5B | 2.33 | 271.77 | 0.84 | 0.54 |
| | WeTok-AR-XL (Ours) | 1.5B | 2.31 | 276.55 | 0.84 | 0.55 |

tokenizer. More details in Sup. E.2. As shown in Tab. 5, our WeTok-based AR model achieves state-of-the-art performance on the ImageNet 50K validation set. This result demonstrates that WeTok is a effective tokenizer not only for image reconstruction but also for high-fidelity visual generation. As shown in Fig. 15, we show the realistic and diverse image generation results of our WeTok-AR-XL.

## 5 CONCLUSION

In this paper, we introduce WeTok, a family of powerful discrete visual tokenizer designed to resolve the conflict between compression ratio and reconstruction. We propose GQ to provides a scalable and memory-efficient solution for codebooks, and GD that excels at producing high-fidelity images even from highly compressed representations. Through extensive experiments, we demonstrated that WeTok consistently outperforms existing state-of-the-art discrete and continuous tokenizers in both in-distribution and zero-shot reconstruction tasks across a wide range of compression ratios. Furthermore, by integrating WeTok into an autoregressive framework, we achieved state-of-the-art performance in class-conditional image generation, confirming that its learned tokens are highly effective for downstream generative tasks. WeTok proves that discrete tokenizers can achieve superior reconstruction quality without compromising their inherent advantage in compression.

## ACKNOWLEDGMENTS

This work was supported by Guangdong Science and Technology Program (Grant No. 2024TQ08X365).

## ETHICS STATEMENT

Our WeTok is a family of powerful discrete visual tokenizer designed to resolve the conflict between compression ratio and reconstruction. To ensure ethical compliance, our training data is carefully selected from public sources and take deliberate measures to minimize potential biases, fully aligning with universal ethical guidelines. We explicitly emphasize that this framework is not intended for misuse in achieving harmful purposes; downstream users are encouraged to adhere to ethical principles when applying the technology. Additionally, all authors declare no conflicts of interest related to this work.

## REPEATABILITY STATEMENT

To ensure full reproducibility, we will publicly release all code and data necessary to replicate our experiments. Comprehensive implementation details, including model architecture, hyperparameters, and training methodology, are provided in this paper and its appendix. We are committed to open-sourcing all essential resources to ensure that our findings can be fully verified and built upon by the research community.

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

# A  THEORETICAL ANALYSIS ON APPROXIMATION ERRORS

## A.1  ABSTRACT ALGEBRA MODELING OF THE GROUPING OF THE GQ METHOD

To rigorously analyze the relationships between different grouping strategies within the Group Quantization (GQ) framework, we employ concepts from abstract algebra, specifically lattice theory. This formalization allows us to create a structured understanding of how different groupings relate to one another in terms of granularity. We begin by defining the set of all possible grouping configurations.

Let the latent feature be represented by a sequence of $d$ elements, indexed by the set $S = \{1, 2, \ldots, d\}$.

**Definition A.1.** *(**The Set of Groupings** $G$) A grouping $g \in G$ of the set of indices $S$ is a collection of non-empty, disjoint, and contiguous blocks whose union is $S$. A block is a set of consecutive integers. For any $g \in G$, if we order its blocks $B_1, B_2, \ldots, B_k$ such that for any $i < j$, every element in $B_i$ is smaller than every element in $B_j$, then this grouping is unique.*

**Remark A.2.** *The cardinality of the grouping set $G$ is $2^{d-1}$. This is because we can think of placing a divider in any of the $d - 1$ spaces between the elements $\{1, 2\}, \{2, 3\}, \ldots, \{d - 1, d\}$. For each space, we can either place a divider or not, leading to $2^{d-1}$ possible groupings.*

Having established the universe of all possible groupings, we now introduce a relation to formally compare them based on their level of subdivision.

**Definition A.3.** *(**The Refinement Ordering Relation** $\preceq$) Let $g_1$ and $g_2$ be two groupings in $G$. We say that $g_1 \preceq g_2$ if $g_2$ is a refinement of $g_1$. This means that every block in $g_2$ is a subset of some block in $g_1$.*

**Remark A.4.** *For example, let $g_1 = \{\{1, 2, 3\}, \{4\}\}$ and $g_2 = \{\{1\}, \{2, 3\}, \{4\}\}$. Here, $\{1\} \subseteq \{1, 2, 3\}$, $\{2, 3\} \subseteq \{1, 2, 3\}$, and $\{4\} \subseteq \{4\}$. Therefore, $g_1 \preceq g_2$.*

This refinement relation imposes a formal structure on the set $G$. We first demonstrate that this structure satisfies the fundamental properties of a partially ordered set.

**Lemma A.5.** $(G, \preceq)$ *is a partially ordered set.*

*Proof.* A partially ordered set must satisfy three properties: reflexivity, antisymmetry, and transitivity.

- **Reflexivity**: For any $g \in G$, $g \preceq g$. This is true because every block in $g$ is a subset of itself.

- **Antisymmetry**: For any $g_1, g_2 \in G$, if $g_1 \preceq g_2$ and $g_2 \preceq g_1$, then $g_1 = g_2$. If $g_1 \preceq g_2$, every block of $g_2$ is a subset of a block of $g_1$. If $g_2 \preceq g_1$, every block of $g_1$ is a subset of a block of $g_2$. Let $B_2$ be a block in $g_2$. Then $B_2 \subseteq B_1$ for some block $B_1$ in $g_1$. Also, $B_1 \subseteq B_2'$ for some block $B_2'$ in $g_2$. Thus, $B_2 \subseteq B_1 \subseteq B_2'$. Since the blocks of a grouping are disjoint, we must have $B_2 = B_2'$. This implies $B_1 = B_2$. Since this holds for all blocks, $g_1$ and $g_2$ consist of the same blocks, so $g_1 = g_2$.

- **Transitivity**: For any $g_1, g_2, g_3 \in G$, if $g_1 \preceq g_2$ and $g_2 \preceq g_3$, then $g_1 \preceq g_3$. If $g_1 \preceq g_2$, every block in $g_2$ is a subset of some block in $g_1$. If $g_2 \preceq g_3$, every block in $g_3$ is a subset of some block in $g_2$. Let $B_3$ be a block in $g_3$. Then $B_3 \subseteq B_2$ for some block $B_2$ in $g_2$. And $B_2 \subseteq B_1$ for some block $B_1$ in $g_1$. Therefore, $B_3 \subseteq B_1$. Since this is true for every block in $g_3$, it follows that $g_1 \preceq g_3$.

$\square$

The poset $(G, \preceq)$ possesses an even stronger and more useful structure. We show that for any two groupings, we can always find a unique common coarser grouping (meet) and a unique common finer grouping (join), which establishes the structure as a lattice.

**Lemma A.6.** $(G, \preceq)$ *is a Lattice.*

*Proof.* To prove that it is a lattice, we must show that for every pair of elements $g_1, g_2 \in G$, there exists a unique greatest lower bound (GLB or meet) and a unique least upper bound (LUB or join).

- The Greatest Lower Bound (Meet)

    Let $g_1, g_2 \in G$. Their meet, denoted $g_1 \wedge g_2$, is the finest grouping that is still coarser than both $g_1$ and $g_2$. The meet is constructed as follows: the blocks of $g_1 \wedge g_2$ are formed by taking unions of blocks from $g_1$ and $g_2$ that overlap.

    Formally, we can define an equivalence relation $\sim$ on the set of indices $S = \{1, \ldots, d\}$, where $i \sim j$ if and only if $i$ and $j$ belong to the same block in both $g_1$ and $g_2$ after transitively closing the relation (i.e., if $i$ and $k$ are in the same block of $g_1$, and $k$ and $j$ are in the same block of $g_2$, then $i \sim j$). The equivalence classes of this relation form the blocks of a new grouping, which is $g_1 \wedge g_2$.

    By its construction, $g_1 \wedge g_2 \preceq g_1$ and $g_1 \wedge g_2 \preceq g_2$. Any other grouping $g'$ that is also a lower bound ($g' \preceq g_1$ and $g' \preceq g_2$) will be coarser than $g_1 \wedge g_2$, meaning $g' \preceq g_1 \wedge g_2$. Thus, $g_1 \wedge g_2$ is the unique greatest lower bound.

- The Least Upper Bound (Join)

    The join of $g_1$ and $g_2$, denoted $g_1 \vee g_2$, is the coarsest grouping that is a refinement of both $g_1$ and $g_2$. The blocks of the join are formed by the non-empty intersections of the blocks from $g_1$ and $g_2$.

    Formally, for each block $B_{1,i} \in g_1$ and each block $B_{2,j} \in g_2$, we form a new block $B_{1,i} \cap B_{2,j}$. The set of all such non-empty intersections forms the grouping $g_1 \vee g_2$.

    By its construction, every block in $g_1 \vee g_2$ is a subset of a block in $g_1$ and a block in $g_2$, so $g_1 \preceq g_1 \vee g_2$ and $g_2 \preceq g_1 \vee g_2$. Any other grouping $g''$ that is also an upper bound ($g_1 \preceq g''$ and $g_2 \preceq g''$) will be a finer partition than $g_1 \vee g_2$, meaning $g_1 \vee g_2 \preceq g''$. Thus, $g_1 \vee g_2$ is the unique least upper bound.

Since a unique meet and join exist for any pair of elements in $G$, $(G, \preceq)$ is a lattice. $\square$

Finally, we consider the extremal elements of this lattice. These correspond to the LFQ and BSQ quantization strategies.

**Proposition A.7.** $(G, \preceq)$ *is a bounded lattice, and its universal least element is the case of $k = 1$ (corresponding to the case of LFQ). Its universal greatest element is the case of $k = d$ (corresponding to the case of BSQ).*

*Proof.* obvious. $\square$

## A.2 ANALYSIS ON THE APPROXIMATION ERROR

Having established the lattice structure of the grouping strategies, we now analyze how the choice of a specific grouping $g$ affects a defined approximation error $e(g)$. This error measures the difference between a sum of products and a product of sums of spatially-dependent probabilities, which can be interpreted as a measure of the statistical coupling across the spatial dimensions.

**Definition A.8.** *For any $g \in G$, let its approximation error be defined as:*

$$e(g) = \left| \sum_{i=1}^{h} \sum_{j=1}^{w} \prod_{k=1}^{|g|} q_G(\mathbf{c}_k | \mathcal{U}_G[i,j,k]) - \prod_{k=1}^{|g|} \sum_{i=1}^{h} \sum_{j=1}^{w} q_G(\mathbf{c}_k | \mathcal{U}_G[i,j,k]) \right| \tag{11}$$

We now propose that this approximation error behaves monotonically with respect to the refinement ordering relation $\preceq$. Specifically, a finer grouping (one with more blocks) will result in a larger approximation error.

**Proposition A.9.** *Given two groupings $g_1, g_2 \in G$ such that $g_2 \preceq g_1$ (i.e., $g_1$ is a refinement of $g_2$), then under the assumption that the variables associated with each block are independent, the approximation error is non-decreasing with refinement:*

$$e(g_2) \leq e(g_1) \tag{12}$$

*Proof.* To simplify the proof, let's introduce more compact notation.

Let the spatial index $s$ represent the pair $(i, j)$, where $s$ ranges from 1 to $N = hw$. Let $X_{s,k}(g) = q_G(\mathbf{c}_k | \mathcal{U}_G[i,j,k])$ denote the non-negative term for block $k$ of grouping $g$ at spatial location $s$. Where the context is clear, we write $X_{s,k}$. Let $m = |g|$ be the number of blocks in grouping $g$.

The error can now be written as:

$$e(g) = \left| \sum_{s=1}^{N} \prod_{k=1}^{m} X_{s,k} - \prod_{k=1}^{m} \sum_{s=1}^{N} X_{s,k} \right|$$

The product of sums can be expanded as:

$$\prod_{k=1}^{m} \sum_{s=1}^{N} X_{s,k} = \sum_{s_1=1}^{N} \sum_{s_2=1}^{N} \cdots \sum_{s_m=1}^{N} \left( X_{s_1,1} X_{s_2,2} \ldots X_{s_m,m} \right)$$

This summation is over all combinations of spatial indices $(s_1, \ldots, s_m)$. We can split this sum into two parts: one where all indices are identical ($s_1 = \cdots = s_m = s$), and one where the indices are not all identical.

$$\prod_{k=1}^{m} \sum_{s=1}^{N} X_{s,k} = \sum_{s=1}^{N} \left( \prod_{k=1}^{m} X_{s,k} \right) + \sum_{\substack{(s_1, \ldots, s_m) \\ \text{not all equal}}} \left( X_{s_1,1} \ldots X_{s_m,m} \right)$$

Rearranging this gives:

$$\prod_{k=1}^{m} \sum_{s=1}^{N} X_{s,k} - \sum_{s=1}^{N} \prod_{k=1}^{m} X_{s,k} = \sum_{\substack{(s_1, \ldots, s_m) \\ \text{not all equal}}} \left( X_{s_1,1} \ldots X_{s_m,m} \right)$$

Since $q_G(\cdot)$ is the output of a softmax function, each term $X_{s,k}$ is non-negative. Therefore, the sum on the right-hand side is also non-negative. This allows us to drop the absolute value bars.

$$e(g) = \prod_{k=1}^{m} \sum_{s=1}^{N} X_{s,k} - \sum_{s=1}^{N} \prod_{k=1}^{m} X_{s,k} \geq 0$$

The relation $g_2 \preceq g_1$ means that $g_1$ is obtained from $g_2$ by a sequence of one or more block subdivisions. It is sufficient to prove that the error increases after a single subdivision, as the general result follows by repeated application.

Let $|g_2| = m$ and $|g_1| = m + 1$. Per the hint that a finer group has more terms, we model refinement by introducing an additional multiplicative set of variables $\{X_{s,m+1}\}_{s=1}^{N}$ into the error calculation. Let $\Delta_m = e(g_2)$ and $\Delta_{m+1} = e(g_1)$.

$$\Delta_{m+1} = \left( \prod_{k=1}^{m+1} \sum_{s=1}^{N} X_{s,k} \right) - \sum_{s=1}^{N} \left( \prod_{k=1}^{m+1} X_{s,k} \right)$$

We separate the $(m+1)$-th term:

$$\Delta_{m+1} = \left(\prod_{k=1}^{m}\sum_{s=1}^{N} X_{s,k}\right)\left(\sum_{s=1}^{N} X_{s,m+1}\right) - \sum_{s=1}^{N}\left(\prod_{k=1}^{m} X_{s,k}\right) X_{s,m+1}$$

From the definition of $\Delta_m$, we substitute $\prod_{k=1}^{m}\sum_s X_{s,k} = \Delta_m + \sum_s \prod_{k=1}^{m} X_{s,k}$:

$$\Delta_{m+1} = \left(\Delta_m + \sum_{s=1}^{N}\prod_{k=1}^{m} X_{s,k}\right)\left(\sum_{s=1}^{N} X_{s,m+1}\right) - \sum_{s=1}^{N}\left(\prod_{k=1}^{m} X_{s,k}\right) X_{s,m+1}$$

Distributing the terms yields the recurrence relation:

$$\Delta_{m+1} = \Delta_m \left(\sum_{s=1}^{N} X_{s,m+1}\right) + \left[\left(\sum_{s=1}^{N}\prod_{k=1}^{m} X_{s,k}\right)\left(\sum_{s=1}^{N} X_{s,m+1}\right) - \sum_{s=1}^{N}\left(\left(\prod_{k=1}^{m} X_{s,k}\right) X_{s,m+1}\right)\right]$$

The expression is a sum of two components.

1. The first component is $\Delta_m \left(\sum_{s=1}^{N} X_{s,m+1}\right)$. Since $\Delta_m \geq 0$ and $X_{s,m+1} \geq 0$, this term is non-negative.

2. The second component, in brackets, is of the form $(\sum_s A_s)(\sum_s B_s) - \sum_s(A_s B_s)$ where $A_s = \prod_{k=1}^{m} X_{s,k}$ and $B_s = X_{s,m+1}$ are non-negative. As shown in Step 2, this form is always non-negative.

Thus, $\Delta_{m+1}$ is the sum of two non-negative quantities. This leads to the inequality $\Delta_{m+1} \geq \Delta_m \left(\sum_{s=1}^{N} X_{s,m+1}\right)$. To ensure that $\Delta_{m+1} \geq \Delta_m$, we rely on the reasonable assumption that for any block $k$, the total probability mass over the large spatial domain is not contractive, i.e., $\sum_{s=1}^{N} X_{s,k} \geq 1$. With this assumption, we have:

$$\Delta_{m+1} \geq \Delta_m$$

We have shown that for a single refinement step that increases the number of blocks from $m$ to $m+1$, the error is non-decreasing. Since any refinement $g_1$ of $g_2$ corresponds to a sequence of such steps, the error for the finer grouping $g_1$ will be greater than or equal to the error for the coarser grouping $g_2$.

$$e(g_1) \geq e(g_2)$$

This proves the proposition. $\qquad\square$

Then, the Proposition 3.1 in the main paper is a direct result of Proposition A.7 and A.9.

## B    MORE RELATED WORK

### B.1    CONTINUOUS TOKENIZER

Generative modeling in the pixel space typically requires extensive compute resources (Chen et al., 2020a; Ho et al., 2020). Subsequent works (Rombach et al., 2022b; Podell et al., 2023; Peebles & Xie, 2023; Esser et al., 2024b; Batifol et al., 2025; Zhuang et al., 2025; 2024; Esser et al., 2024a; Zha et al., 2025; Chen et al., 2025a) adopt VAE (Kingma & Welling, 2013), which projects visual content from pixels to latent features, achieving efficient and photo-realistic visual generation at high resolution. FLUX-VAE (Batifol et al., 2025) shows the state-of-the-art performance in both reconstruction quality and generalization ability across all continuous tokenizers. However, continuous tokenizer is criticized for its low compression rate, because latent features are usually stored and calculated in `float32` or `bfloat16`. Therefore, discrete tokenizers that can store data in `int` or `bool` seem to be more promising in terms of compression capabilities.

## B.2 Discrete Tokenizer

VQVAE (Van Den Oord et al., 2017) and VQGAN (Esser et al., 2021) employ vector-quantization (VQ) to transform visual input into discrete tokens. But they both suffer from low reconstruction quality caused by instability of the codebook utilization. To overcome these drawbacks, one line of work introduces specific optimization strategies or modules to improve performance (Lee et al., 2022b; Shi et al., 2024; Zhu et al., 2024; Yu et al., 2024c). Another line of work focuses on mitigating the training instability when scaling up the codebook size by using grouped codebooks (Ma et al., 2025; Jia et al., 2025; Zhang et al., 2025). These methods split the input feature into groups along the channel dimension, where each group is then looked up using a sub-codebook. However, VQ-based tokenizers still introduce additional inference and training costs due to the lookup operation (Yu et al., 2021b; Lee et al., 2022b; Fang et al., 2025). MAGVIT-v2 (Yu et al., 2024a) introduces Lookup-Free Quantization (LFQ) to address this extra cost and proposes the entropy loss (Chang et al., 2022; Jansen et al., 2019) to ensure the utilization of the codebook. However, the entropy loss causes unaffordable memory cost as it scales linearly with the codebook, limiting the further expansion of the codebook. BSQ (Zhao et al., 2024b) is proposed to mitigate this issue by assuming independence between the bits of the binary code, while this strong assumption leads to performance degradation. In contrast, WeTok does not rely on explicit codebooks, and eliminate the memory usage caused by entropy loss while having better performance than LFQ.

## B.3 Autoregressive Visual Generation

The autoregressive (AR) modeling paradigm, which underpins modern Large Language Models (LLMs) (Vaswani et al., 2017), has been successfully adapted for visual generation (Chen et al., 2020a), where models learn to predict sequences of discrete tokens for images (Ramesh et al., 2021b; Ding et al., 2021; Liu et al., 2024) and videos (Hong et al., 2022; Kondratyuk et al., 2023). Recent AR models (Sun et al., 2024; Team, 2024; Wu et al., 2025a; Wang et al., 2024b; Liu et al., 2025) achieve remarkable image quality, highlighting the potential of this paradigm. Notably, the AR models is critically dependent on the visual tokenizer. Therefore, we adopt our WeTok to the AR (Sun et al., 2024) framework to enable high-fidelity autoregressive generation. This shows that WeTok is not only capable of compression, but its compressed features are also suitable for generative models.

## C Iteration invariance of WeTok

A common use case in practice involves compressing images for transmission and subsequently decompressing them upon reception. However, this process is often iterative, where an image may undergo multiple cycles of compression and decompression (Joshi et al., 2000). While modern tokenizers have proven effective for image compression and even show potential as next-generation compression algorithms, an unexpected issue arises. As illustrated in Fig. 11 and 12, we observe that when state-of-the-art continuous tokenizers (Esser et al., 2024b; Labs, 2024) are used for compression-decompression iterations, the image quality progressively degrades with each iteration. Accordingly, we proceeded to quantitatively analyze the cosine similarity between the latent representation of the original image and that of the image after undergoing multiple compression-decompression iterations. As illustrated in Fig. 13, we selected WeTok trained on general-domain data with a compression rate of 192 for our evaluation. The results demonstrate that after multiple iterations, the image processed by WeTok gradually converges to a stable state. In contrast, the images processed by FLUX-VAE and SD-VAE 3.5 collapse.

## D More Result

### D.1 Comparison with state-of-the-Art

**Visual Reconstruction.** As shown in Fig. 14.

**Visual Generation.** As shown in Fig. 15 and Tab. 6.

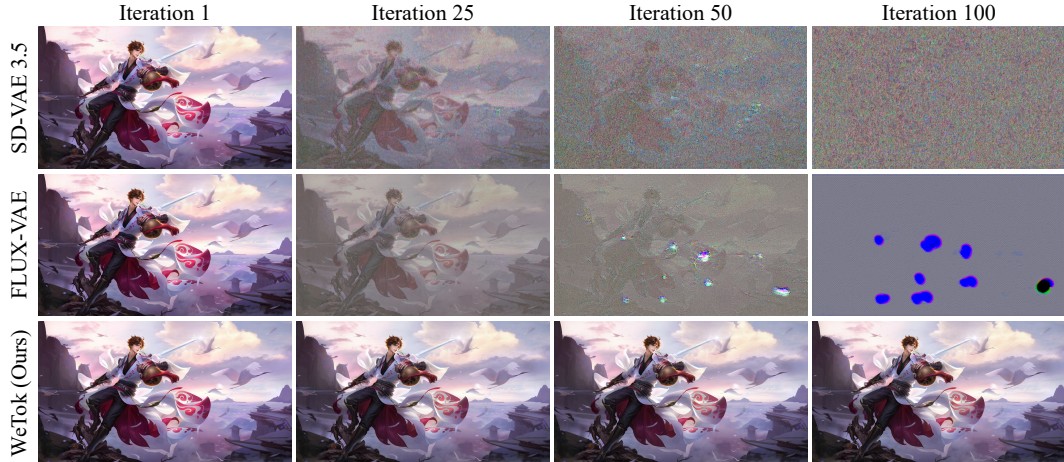

Figure 11: **Qualitative comparison of *image_1* on compression-decompression iteration.**

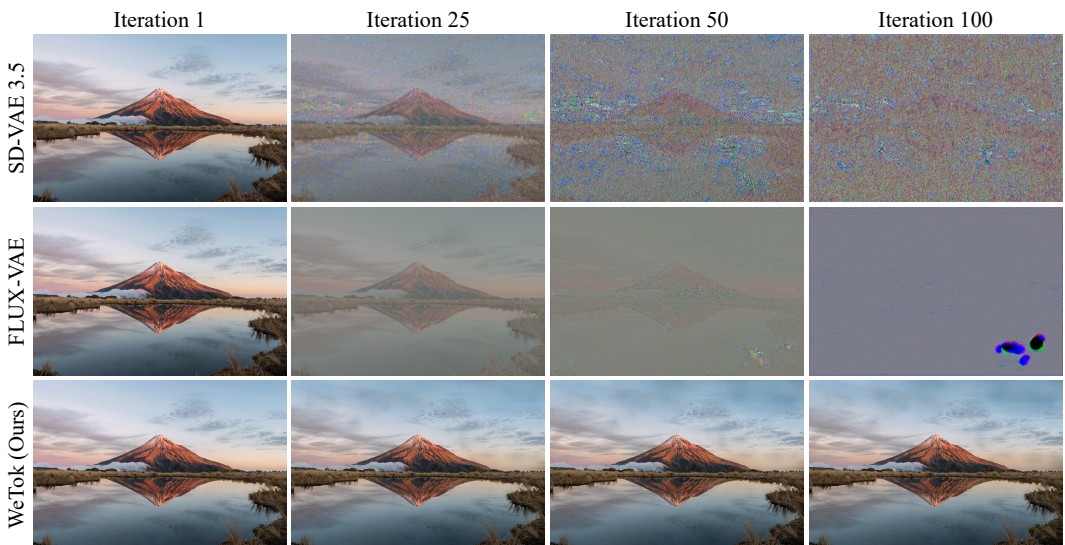

Figure 12: **Qualitative comparison of *image_2* on compression-decompression iteration.**

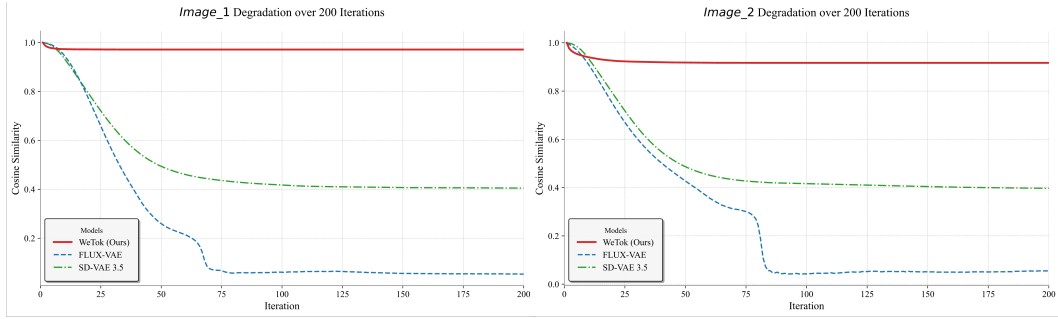

Figure 13: **Quantitative comparison of *image_1* and *image_2* on compression-decompression iteration.** After multiple iterations, the images processed by WeTok gradually converged to a stable value, while the images processed by FLUX-VAE and SD-VAE 3.5 collapse.

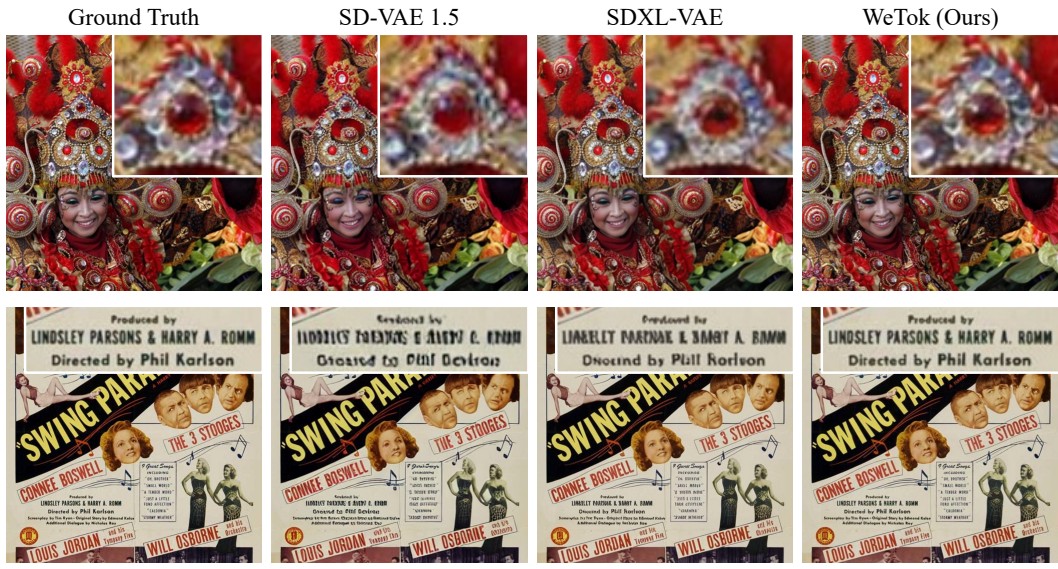

Figure 14: **More qualitative comparison of $512 \times 512$ image reconstruction on TokBench.**

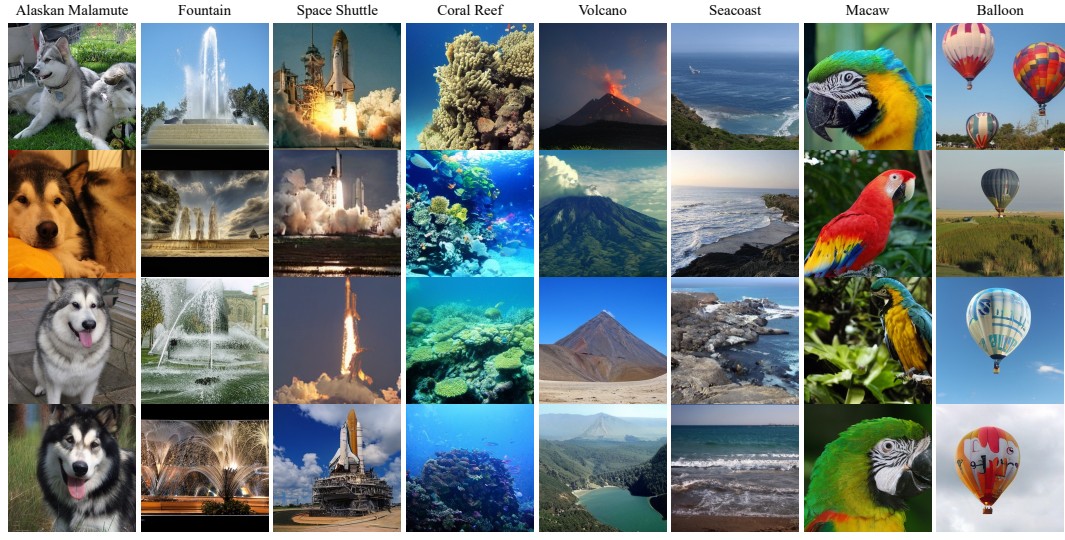

Figure 15: **More WeTok-AR-XL generated samples at $256 \times 256$ resolution.**

Table 6: **Class-conditional generation on $256 \times 256$ ImageNet.** $*$ specifies the generated images are $384 \times 384$ and are resized to 256×256 for evaluation.

| Type | Model | #Para. | FID↓ | IS↑ | Precision↑ | Recall↑ |
|---|---|---|---|---|---|---|
| GAN | BigGAN (Brock et al., 2018) | 112M | 6.95 | 224.5 | 0.89 | 0.38 |
| | GigaGAN (Kang et al., 2023) | 569M | 3.45 | 225.5 | 0.84 | 0.61 |
| | StyleGan-XL (Sauer et al., 2022) | 166M | 2.30 | 265.1 | 0.78 | 0.53 |
| Diffusion | ADM (Dhariwal & Nichol, 2021) | 554M | 10.94 | 101.0 | 0.69 | 0.63 |
| | CDM (Ho et al., 2021) | − | 4.88 | 158.7 | − | − |
| | LDM-4 (Rombach et al., 2022b) | 400M | 3.60 | 247.7 | − | − |
| | DiT-XL/2 (Peebles & Xie, 2022) | 675M | 2.27 | 278.2 | 0.83 | 0.57 |
| | SiT-XL/2 (Ma et al., 2024) | 675M | 2.06 | 277.50 | 0.83 | 0.59 |
| Mask. | MaskGIT (Chang et al., 2022) | 227M | 6.18 | 182.1 | 0.80 | 0.51 |
| | MaskGIT-re (Chang et al., 2022) | 227M | 4.02 | 355.6 | − | − |
| VAR | VAR-d16 (Tian et al., 2024) | 310M | 3.30 | 274.4 | 0.84 | 0.51 |
| | VAR-d20 (Tian et al., 2024) | 600M | 2.57 | 302.6 | 0.83 | 0.56 |
| | VAR-d24 (Tian et al., 2024) | 1.0B | 2.09 | 312.9 | 0.82 | 0.59 |
| | VAR-d30 (Tian et al., 2024) | 2.0B | 1.92 | 323.1 | 0.82 | 0.59 |
| AR | VQGAN (Esser et al., 2020) | 227M | 18.65 | 80.4 | 0.78 | 0.26 |
| | VQGAN (Esser et al., 2020) | 1.4B | 15.78 | 74.3 | − | − |
| | VQGAN-re (Esser et al., 2020) | 1.4B | 5.20 | 280.3 | − | − |
| | ViT-VQGAN (Yu et al., 2021a) | 1.7B | 4.17 | 175.1 | − | − |
| | ViT-VQGAN-re (Yu et al., 2021a) | 1.7B | 3.04 | 227.4 | − | − |
| | RQTran. (Lee et al., 2022a) | 3.8B | 7.55 | 134.0 | − | − |
| | RQTran.-re (Lee et al., 2022a) | 3.8B | 3.80 | 323.7 | − | − |
| | LlamaGen-L$^*$ (Sun et al., 2024) | 343M | 3.07 | 256.06 | 0.83 | 0.52 |
| | LlamaGen-XL$^*$ (Sun et al., 2024) | 775M | 2.62 | 244.08 | 0.80 | 0.57 |
| | LlamaGen-XXL$^*$ (Sun et al., 2024) | 1.4B | 2.34 | 253.90 | 0.80 | 0.59 |
| | LlamaGen-L (Sun et al., 2024) | 343M | 3.80 | 248.28 | 0.83 | 0.51 |
| | LlamaGen-XL (Sun et al., 2024) | 775M | 3.39 | 227.08 | 0.81 | 0.54 |
| | LlamaGen-XXL (Sun et al., 2024) | 1.4B | 3.09 | 253.61 | 0.83 | 0.53 |
| | UniTok (Ma et al., 2025) | 1.4B | 2.51 | 216.7 | 0.82 | 0.57 |
| | Open-MAGVIT2-AR-B (Luo et al., 2024) | 343M | 3.08 | 258.26 | 0.85 | 0.51 |
| | Open-MAGVIT2-AR-L (Luo et al., 2024) | 804M | 2.51 | 271.70 | 0.84 | 0.54 |
| | Open-MAGVIT2-AR-XL (Luo et al., 2024) | 1.5B | 2.33 | 271.77 | 0.84 | 0.54 |
| | WeTok-AR-XL (Ours) | 1.5B | 2.31 | 276.55 | 0.84 | 0.55 |

# E  MORE IMPLEMENTATION DETAILS

## E.1  ABLATION STUDY

**Quantization method.** As shown in Tab. 7, 8 and 9.

**Generative decoder.** As shown in Tab. 10 and 11.

**Number of group in GQ.** As shown in Tab. 12, 13, 14, 15, 16 and 17.

**Model architecture.** As shown in Tab. 18, 19, 20, 21, 22, 23, 24, 25, and 26.

**Training data.** As shown in Tab. 27 and 28.

**Learning rate schedule.** As shown in Tab. 29.

**Parameter of autoregressive model.** As shown in Tab. 30, 31 and 32.

## E.2  COMPARISON WITH STATE-OF-THE-ART

**Visual Reconstruction.** The settings of in-distribution comparison are shown in Tab. 33 and 34. The settings of general-domain comparison are shown in Tab. 35, 36, 37, 38 and 39, .

**Visual Generation.** As shown in Tab. 32.

Table 7: **GQ training setting.**

| config | GQ |
|---|---|
| training data | IN-1K training set |
| image size | [256, 256] |
| data augmentation | random crop |
| downsample | $16 \times 16$ |
| ema | True |
| $g$ (group number) | 2 |
| $d'$ (group channel) | 8 |
| optimizer | Adam |
| optimizer momentum | $\beta_1, \beta_2 = 0.5, 0.9$ |
| weight decay | 0 |
| learning rate schedule | consistent |
| learning rate | 1e-4 |
| warmup steps | 0 |
| cos decay end ratio | 1 |
| total steps | 250250 |
| channel_mult | [1,1,2,2,4] |
| channel | 128 |
| num_res_blocks | 4 |
| generative decoder | False |
| per GPU batchsize | 16 |
| global batchsize | 128 |
| GPU number | 8 H20 |

Table 8: **LFQ training setting.**

| config | LFQ |
|---|---|
| training data | IN-1K training set |
| image size | [256, 256] |
| data augmentation | random crop |
| downsample | $16 \times 16$ |
| ema | True |
| $g$ (group number) | 1 |
| $d'$ (group channel) | 16 |
| optimizer | Adam |
| optimizer momentum | $\beta_1, \beta_2 = 0.5, 0.9$ |
| weight decay | 0 |
| learning rate schedule | consistent |
| learning rate | 1e-4 |
| warmup steps | 0 |
| cos decay end ratio | 1 |
| total steps | 250250 |
| channel_mult | [1,1,2,2,4] |
| channel | 128 |
| num_res_blocks | 4 |
| generative decoder | False |
| per GPU batchsize | 16 |
| global batchsize | 128 |
| GPU number | 8 H20 |

Table 9: **BSQ training setting.**

| config | BSQ |
|---|---|
| training data | IN-1K training set |
| image size | [256, 256] |
| data augmentation | random crop |
| downsample | $16 \times 16$ |
| ema | True |
| $g$ (group number) | 16 |
| $d'$ (group channel) | 1 |
| optimizer | Adam |
| optimizer momentum | $\beta_1, \beta_2 = 0.5, 0.9$ |
| weight decay | 0 |
| learning rate schedule | consistent |
| learning rate | 1e-4 |
| warmup steps | 0 |
| cos decay end ratio | 1 |
| total steps | 250250 |
| channel_mult | [1,1,2,2,4] |
| channel | 128 |
| num_res_blocks | 4 |
| generative decoder | False |
| per GPU batchsize | 16 |
| global batchsize | 128 |
| GPU number | 8 H20 |

Table 10: **Stage-1 training setting.**

| config | Stage-1 |
|---|---|
| training data | general domain dataset |
| image size | [256, 256] |
| data augmentation | random crop |
| downsample | $32 \times 32$ |
| ema | True |
| $g$ (group number) | 4 |
| $d'$ (group channel) | 8 |
| optimizer | Adam |
| optimizer momentum | $\beta_1, \beta_2 = 0.5, 0.9$ |
| weight decay | 0 |
| learning rate schedule | consistent |
| learning rate | 1e-4 |
| warmup steps | 0 |
| cos decay end ratio | 1 |
| total steps | 550550 |
| channel_mult | [1,1,2,2,4,8] |
| channel | 256 |
| num_res_blocks | 4 |
| generative decoder | False |
| per GPU batchsize | 6 |
| global batchsize | 1056 |
| GPU number | 176 H20 |

Table 11: **Stage-2 training setting.**

| config | Stage-2 |
|---|---|
| training data | general domain dataset |
| image size | [256, 256] |
| data augmentation | random crop |
| downsample | $32 \times 32$ |
| ema | True |
| $g$ (group number) | 4 |
| $d'$ (group channel) | 8 |
| optimizer | Adam |
| optimizer momentum | $\beta_1, \beta_2 = 0.5, 0.9$ |
| weight decay | 0 |
| learning rate schedule | consistent |
| learning rate | 1e-4 |
| warmup steps | 0 |
| cos decay end ratio | 1 |
| total steps | 550550 |
| channel_mult | [1,1,2,2,4,8] |
| channel | 256 |
| num_res_blocks | 4 |
| generative decoder | True |
| per GPU batchsize | 6 |
| global batchsize | 1056 |
| GPU number | 176 H20 |

Table 12: **1 group training setting.**

| config | 1 group |
|---|---|
| training data | IN-1K training set |
| image size | [256, 256] |
| data augmentation | random crop |
| downsample | $16 \times 16$ |
| ema | True |
| $g$ (group number) | 1 |
| $d'$ (group channel) | 8 |
| optimizer | Adam |
| optimizer momentum | $\beta_1, \beta_2 = 0.5, 0.9$ |
| weight decay | 0 |
| learning rate schedule | consistent |
| learning rate | 1e-4 |
| warmup steps | 0 |
| cos decay end ratio | 1 |
| total steps | 250250 |
| channel_mult | [1,1,2,2,4] |
| channel | 128 |
| num_res_blocks | 4 |
| generative decoder | False |
| per GPU batchsize | 16 |
| global batchsize | 128 |
| GPU number | 8 H20 |

Table 13: **2 group training setting.**

| config | 2 group |
|---|---|
| training data | IN-1K training set |
| image size | [256, 256] |
| data augmentation | random crop |
| downsample | $16 \times 16$ |
| ema | True |
| $g$ (group number) | 2 |
| $d'$ (group channel) | 8 |
| optimizer | Adam |
| optimizer momentum | $\beta_1, \beta_2 = 0.5, 0.9$ |
| weight decay | 0 |
| learning rate schedule | consistent |
| learning rate | 1e-4 |
| warmup steps | 0 |
| cos decay end ratio | 1 |
| total steps | 250250 |
| channel_mult | [1,1,2,2,4] |
| channel | 128 |
| num_res_blocks | 4 |
| generative decoder | False |
| per GPU batchsize | 16 |
| global batchsize | 128 |
| GPU number | 8 H20 |

Table 14: **2 group training setting.**

| config | 4 group |
|---|---|
| training data | IN-1K training set |
| image size | [256, 256] |
| data augmentation | random crop |
| downsample | $16 \times 16$ |
| ema | True |
| $g$ (group number) | 4 |
| $d'$ (group channel) | 8 |
| optimizer | Adam |
| optimizer momentum | $\beta_1, \beta_2 = 0.5, 0.9$ |
| weight decay | 0 |
| learning rate schedule | consistent |
| learning rate | 1e-4 |
| warmup steps | 0 |
| cos decay end ratio | 1 |
| total steps | 250250 |
| channel_mult | [1,1,2,2,4] |
| channel | 128 |
| num_res_blocks | 4 |
| generative decoder | False |
| per GPU batchsize | 16 |
| global batchsize | 128 |
| GPU number | 8 H20 |

Table 15: **8 group training setting.**

| config | 8 group |
|---|---|
| training data | IN-1K training set |
| image size | [256, 256] |
| data augmentation | random crop |
| downsample | $16 \times 16$ |
| ema | True |
| $g$ (group number) | 8 |
| $d'$ (group channel) | 8 |
| optimizer | Adam |
| optimizer momentum | $\beta_1, \beta_2 = 0.5, 0.9$ |
| weight decay | 0 |
| learning rate schedule | consistent |
| learning rate | 1e-4 |
| warmup steps | 0 |
| cos decay end ratio | 1 |
| total steps | 250250 |
| channel_mult | [1,1,2,2,4] |
| channel | 128 |
| num_res_blocks | 4 |
| generative decoder | False |
| per GPU batchsize | 16 |
| global batchsize | 128 |
| GPU number | 8 H20 |

Table 16: **16 group training setting.**

| config | 16 group |
|---|---|
| training data | IN-1K training set |
| image size | [256, 256] |
| data augmentation | random crop |
| downsample | $16 \times 16$ |
| ema | True |
| $g$ (group number) | 16 |
| $d'$ (group channel) | 8 |
| optimizer | Adam |
| optimizer momentum | $\beta_1, \beta_2 = 0.5, 0.9$ |
| weight decay | 0 |
| learning rate schedule | consistent |
| learning rate | 1e-4 |
| warmup steps | 0 |
| cos decay end ratio | 1 |
| total steps | 250250 |
| channel_mult | [1,1,2,2,4] |
| channel | 128 |
| num_res_blocks | 4 |
| generative decoder | False |
| per GPU batchsize | 16 |
| global batchsize | 128 |
| GPU number | 8 H20 |

Table 17: **32 group training setting.**

| config | 32 group |
|---|---|
| training data | IN-1K training set |
| image size | [256, 256] |
| data augmentation | random crop |
| downsample | $16 \times 16$ |
| ema | True |
| $g$ (group number) | 32 |
| $d'$ (group channel) | 8 |
| optimizer | Adam |
| optimizer momentum | $\beta_1, \beta_2 = 0.5, 0.9$ |
| weight decay | 0 |
| learning rate schedule | consistent |
| learning rate | 1e-4 |
| warmup steps | 0 |
| cos decay end ratio | 1 |
| total steps | 250250 |
| channel_mult | [1,1,2,2,4] |
| channel | 128 |
| num_res_blocks | 4 |
| generative decoder | False |
| per GPU batchsize | 16 |
| global batchsize | 128 |
| GPU number | 8 H20 |

Table 18: **128 channel 4 block training setting.**

| config | 128 channel 4 block |
|---|---|
| training data | IN-1K training set |
| image size | [256, 256] |
| data augmentation | random crop |
| downsample | $16 \times 16$ |
| ema | True |
| $g$ (group number) | 4 |
| $d'$ (group channel) | 8 |
| optimizer | Adam |
| optimizer momentum | $\beta_1, \beta_2$=0.5, 0.9 |
| weight decay | 0 |
| learning rate schedule | consistent |
| learning rate | 1e-4 |
| warmup steps | 0 |
| cos decay end ratio | 1 |
| total steps | 250250 |
| channel_mult | [1,1,2,2,4] |
| channel | 128 |
| num_res_blocks | 4 |
| generative decoder | False |
| per GPU batchsize | 16 |
| global batchsize | 128 |
| GPU number | 8 H20 |

Table 19: **192 channel 4 block training setting.**

| config | 192 channel 4 block |
|---|---|
| training data | IN-1K training set |
| image size | [256, 256] |
| data augmentation | random crop |
| downsample | $16 \times 16$ |
| ema | True |
| $g$ (group number) | 4 |
| $d'$ (group channel) | 8 |
| optimizer | Adam |
| optimizer momentum | $\beta_1, \beta_2$=0.5, 0.9 |
| weight decay | 0 |
| learning rate schedule | consistent |
| learning rate | 1e-4 |
| warmup steps | 0 |
| cos decay end ratio | 1 |
| total steps | 250250 |
| channel_mult | [1,1,2,2,4] |
| channel | 192 |
| num_res_blocks | 4 |
| generative decoder | False |
| per GPU batchsize | 8 |
| global batchsize | 128 |
| GPU number | 16 H20 |

Table 20: **256 channel 4 block training setting.**

| config | 256 channel 4 block |
|---|---|
| training data | IN-1K training set |
| image size | [256, 256] |
| data augmentation | random crop |
| downsample | $16 \times 16$ |
| ema | True |
| $g$ (group number) | 4 |
| $d'$ (group channel) | 8 |
| optimizer | Adam |
| optimizer momentum | $\beta_1, \beta_2$=0.5, 0.9 |
| weight decay | 0 |
| learning rate schedule | consistent |
| learning rate | 1e-4 |
| warmup steps | 0 |
| cos decay end ratio | 1 |
| total steps | 250250 |
| channel_mult | [1,1,2,2,4] |
| channel | 256 |
| num_res_blocks | 4 |
| generative decoder | False |
| per GPU batchsize | 4 |
| global batchsize | 128 |
| GPU number | 32 H20 |

Table 21: **128 channel 8 block training setting.**

| config | 128 channel 8 block |
|---|---|
| training data | IN-1K training set |
| image size | [256, 256] |
| data augmentation | random crop |
| downsample | $16 \times 16$ |
| ema | True |
| $g$ (group number) | 4 |
| $d'$ (group channel) | 8 |
| optimizer | Adam |
| optimizer momentum | $\beta_1, \beta_2$=0.5, 0.9 |
| weight decay | 0 |
| learning rate schedule | consistent |
| learning rate | 1e-4 |
| warmup steps | 0 |
| cos decay end ratio | 1 |
| total steps | 250250 |
| channel_mult | [1,1,2,2,4] |
| channel | 128 |
| num_res_blocks | 8 |
| generative decoder | False |
| per GPU batchsize | 8 |
| global batchsize | 128 |
| GPU number | 16 H20 |

Table 22: **128 channel 16 block training setting.**

| config | 128 channel 16 block |
|---|---|
| training data | IN-1K training set |
| image size | [256, 256] |
| data augmentation | random crop |
| downsample | $16 \times 16$ |
| ema | True |
| $g$ (group number) | 4 |
| $d'$ (group channel) | 8 |
| optimizer | Adam |
| optimizer momentum | $\beta_1, \beta_2$=0.5, 0.9 |
| weight decay | 0 |
| learning rate schedule | consistent |
| learning rate | 1e-4 |
| warmup steps | 0 |
| cos decay end ratio | 1 |
| total steps | 250250 |
| channel_mult | [1,1,2,2,4] |
| channel | 128 |
| num_res_blocks | 16 |
| generative decoder | False |
| per GPU batchsize | 4 |
| global batchsize | 128 |
| GPU number | 32 H20 |

Table 23: **192 channel 8 block training setting.**

| config | 192 channel 8 block |
|---|---|
| training data | IN-1K training set |
| image size | [256, 256] |
| data augmentation | random crop |
| downsample | $16 \times 16$ |
| ema | True |
| $g$ (group number) | 4 |
| $d'$ (group channel) | 8 |
| optimizer | Adam |
| optimizer momentum | $\beta_1, \beta_2$=0.5, 0.9 |
| weight decay | 0 |
| learning rate schedule | consistent |
| learning rate | 1e-4 |
| warmup steps | 0 |
| cos decay end ratio | 1 |
| total steps | 250250 |
| channel_mult | [1,1,2,2,4] |
| channel | 192 |
| num_res_blocks | 8 |
| generative decoder | False |
| per GPU batchsize | 4 |
| global batchsize | 128 |
| GPU number | 32 H20 |

Table 24: **256 channel 2 block training setting.**

| config | 256 channel 2 block |
|---|---|
| training data | IN-1K training set |
| image size | [256, 256] |
| data augmentation | random crop |
| downsample | $16 \times 16$ |
| ema | True |
| $g$ (group number) | 4 |
| $d'$ (group channel) | 8 |
| optimizer | Adam |
| optimizer momentum | $\beta_1, \beta_2$=0.5, 0.9 |
| weight decay | 0 |
| learning rate schedule | consistent |
| learning rate | 1e-4 |
| warmup steps | 0 |
| cos decay end ratio | 1 |
| total steps | 250250 |
| channel_mult | [1,1,2,2,4] |
| channel | 256 |
| num_res_blocks | 2 |
| generative decoder | False |
| per GPU batchsize | 8 |
| global batchsize | 128 |
| GPU number | 16 H20 |

Table 25: **384 channel 4 block training setting.**

| config | 384 channel 4 block |
|---|---|
| training data | IN-1K training set |
| image size | [256, 256] |
| data augmentation | random crop |
| downsample | $16 \times 16$ |
| ema | True |
| $g$ (group number) | 4 |
| $d'$ (group channel) | 8 |
| optimizer | Adam |
| optimizer momentum | $\beta_1, \beta_2$=0.5, 0.9 |
| weight decay | 0 |
| learning rate schedule | consistent |
| learning rate | 1e-4 |
| warmup steps | 0 |
| cos decay end ratio | 1 |
| total steps | 250250 |
| channel_mult | [1,1,2,2,4] |
| channel | 384 |
| num_res_blocks | 4 |
| generative decoder | False |
| per GPU batchsize | 2 |
| global batchsize | 128 |
| GPU number | 64 H20 |

Table 26: **256 channel 8 block training setting.**

| config | 256 channel 8 block |
|---|---|
| training data | IN-1K training set |
| image size | [256, 256] |
| data augmentation | random crop |
| downsample | $16 \times 16$ |
| ema | True |
| $g$ (group number) | 4 |
| $d'$ (group channel) | 8 |
| optimizer | Adam |
| optimizer momentum | $\beta_1, \beta_2$=0.5, 0.9 |
| weight decay | 0 |
| learning rate schedule | consistent |
| learning rate | 1e-4 |
| warmup steps | 0 |
| cos decay end ratio | 1 |
| total steps | 250250 |
| channel_mult | [1,1,2,2,4] |
| channel | 256 |
| num_res_blocks | 8 |
| generative decoder | False |
| per GPU batchsize | 2 |
| global batchsize | 128 |
| GPU number | 64 H20 |

Table 27: **ImageNet training setting.**

| config | 1.2M |
|---|---|
| training data | IN-1K training set |
| image size | [256, 256] |
| data augmentation | random crop |
| downsample | $16 \times 16$ |
| ema | True |
| $g$ (group number) | 4 |
| $d'$ (group channel) | 8 |
| optimizer | Adam |
| optimizer momentum | $\beta_1, \beta_2 = 0.5, 0.9$ |
| weight decay | 0 |
| learning rate schedule | consistent |
| learning rate | 1e-4 |
| warmup steps | 0 |
| cos decay end ratio | 1 |
| total steps | 550550 |
| channel_mult | [1,1,2,2,4] |
| channel | 128 |
| num_res_blocks | 4 |
| generative decoder | False |
| per GPU batchsize | 16 |
| global batchsize | 128 |
| GPU number | 8 H20 |

Table 28: **General-domain training setting.**

| config | 400M |
|---|---|
| training data | general domain dataset |
| image size | [256, 256] |
| data augmentation | random crop |
| downsample | $16 \times 16$ |
| ema | True |
| $g$ (group number) | 4 |
| $d'$ (group channel) | 8 |
| optimizer | Adam |
| optimizer momentum | $\beta_1, \beta_2 = 0.5, 0.9$ |
| weight decay | 0 |
| learning rate schedule | consistent |
| learning rate | 1e-4 |
| warmup steps | 0 |
| cos decay end ratio | 1 |
| total steps | 550550 |
| channel_mult | [1,1,2,2,4] |
| channel | 128 |
| num_res_blocks | 4 |
| generative decoder | False |
| per GPU batchsize | 16 |
| global batchsize | 128 |
| GPU number | 8 H20 |

Table 29: **Warm up + cosine decay learning rate schedule training setting.**

| config | warm up + cosine decay |
|---|---|
| training data | general domain dataset |
| image size | [256, 256] |
| data augmentation | random crop |
| downsample | $16 \times 16$ |
| ema | True |
| $g$ (group number) | 4 |
| $d'$ (group channel) | 8 |
| optimizer | Adam |
| optimizer momentum | $\beta_1, \beta_2 = 0.5, 0.9$ |
| weight decay | 0 |
| learning rate schedule | warm up + cosine decay |
| learning rate | 1e-4 |
| warmup steps | 10000 |
| cos decay end ratio | 0.01 |
| total steps | 550550 |
| channel_mult | [1,1,2,2,4] |
| channel | 128 |
| num_res_blocks | 4 |
| generative decoder | False |
| per GPU batchsize | 16 |
| global batchsize | 128 |
| GPU number | 8 H20 |

Table 30: **LlamaGen Base training setting.**

| config | LlamaGen Base |
|---|---|
| training data | IN-1K training set |
| image size | [256, 256] |
| data augmentation | random crop |
| downsample | $16 \times 16$ |
| ema | False |
| optimizer | AdamW |
| optimizer momentum | $\beta_1, \beta_2$=0.9, 0.95 |
| weight decay | 5e-2 |
| learning rate schedule | warm up + linear decay |
| learning rate | 3e-4 |
| warmup epochs | 6 |
| linear decay end ratio | 0.1 |
| total epochs | 1000 |
| dim | 1024 |
| num_head | 16 |
| trans_layers | 24 |
| cond_dim | 1024 |
| factorized_layers | 2 |
| factorized_k | 4 |
| token_drop | 0.1 |
| residual_drop | 0.1 |
| per GPU batchsize | 64 |
| global batchsize | 3072 |
| GPU number | 48 H20 |

Table 31: **LlamaGen Large training setting.**

| config | LlamaGen Large |
|---|---|
| training data | IN-1K training set |
| image size | [256, 256] |
| data augmentation | random crop |
| downsample | $16 \times 16$ |
| ema | False |
| optimizer | AdamW |
| optimizer momentum | $\beta_1, \beta_2$=0.9, 0.95 |
| weight decay | 5e-2 |
| learning rate schedule | warm up + linear decay |
| learning rate | 3e-4 |
| warmup epochs | 6 |
| linear decay end ratio | 0.1 |
| total epochs | 1000 |
| dim | 1280 |
| num_head | 20 |
| trans_layers | 36 |
| cond_dim | 1280 |
| factorized_layers | 3 |
| factorized_k | 4 |
| token_drop | 0.1 |
| residual_drop | 0.1 |
| per GPU batchsize | 32 |
| global batchsize | 3072 |
| GPU number | 96 H20 |

Table 32: **LlamaGen X-Large training setting.**

| config | LlamaGen X-Large |
|---|---|
| training data | IN-1K training set |
| image size | [256, 256] |
| data augmentation | random crop |
| downsample | $16 \times 16$ |
| ema | False |
| optimizer | AdamW |
| optimizer momentum | $\beta_1, \beta_2$=0.9, 0.95 |
| weight decay | 5e-2 |
| learning rate schedule | warm up + linear decay |
| learning rate | 3e-4 |
| warmup epochs | 6 |
| linear decay end ratio | 0.1 |
| total epochs | 1000 |
| dim | 1536 |
| num_head | 24 |
| trans_layers | 48 |
| cond_dim | 1536 |
| factorized_layers | 4 |
| factorized_k | 4 |
| token_drop | 0.1 |
| residual_drop | 0.1 |
| per GPU batchsize | 16 |
| global batchsize | 3072 |
| GPU number | 192 H20 |

Table 33: **Large scale training on ImageNet training set at 16 downsample ratio.**

| config | IN-1K $16\times$ SOTA |
|---|---|
| training data | IN-1K training set |
| image size | [256, 256] |
| data augmentation | random crop |
| downsample | $16 \times 16$ |
| ema | True |
| $g$ (group number) | 4 |
| $d'$ (group channel) | 8 |
| optimizer | Adam |
| optimizer momentum | $\beta_1, \beta_2$=0.5, 0.9 |
| weight decay | 0 |
| learning rate schedule | consistent |
| learning rate | 1e-4 |
| warmup steps | 0 |
| cos decay end ratio | 1 |
| total steps | 400400 |
| channel_mult | [1,1,2,2,4] |
| channel | 256 |
| num_res_blocks | 4 |
| generative decoder | False |
| per GPU batchsize | 8 |
| global batchsize | 1024 |
| GPU number | 128 H20 |

Table 34: **Large scale training on ImageNet training set at 8 downsample ratio.**

| config | IN-1K $8\times$ SOTA |
|---|---|
| training data | IN-1K training set |
| image size | [256, 256] |
| data augmentation | random crop |
| downsample | $8 \times 8$ |
| ema | True |
| $g$ (group number) | 4 |
| $d'$ (group channel) | 8 |
| optimizer | Adam |
| optimizer momentum | $\beta_1, \beta_2$=0.5, 0.9 |
| weight decay | 0 |
| learning rate schedule | consistent |
| learning rate | 1e-4 |
| warmup steps | 0 |
| cos decay end ratio | 1 |
| total steps | 350350 |
| channel_mult | [1,2,2,4] |
| channel | 256 |
| num_res_blocks | 4 |
| generative decoder | False |
| per GPU batchsize | 8 |
| global batchsize | 1024 |
| GPU number | 128 H20 |

Table 35: **Large scale training on general-domain dataset at 768 compression ratio.**

| config | 768 compression ratio SOTA | | |
|---|---|---|---|
| training data | general domain dataset | | |
| image size | [256, 256] | | |
| data augmentation | random crop | | |
| downsample | $32 \times 32$ | | |
| ema | True | | |
| $g$ (group number) | 4 | | |
| $d'$ (group channel) | 8 | | |
| optimizer | Adam | | |
| optimizer momentum | $\beta_1, \beta_2$=0.5, 0.9 | | |
| weight decay | 0 | | |
| learning rate schedule | consistent | | |
| learning rate | 1e-4 | 1e-4 | 1e-5 |
| warmup steps | 0 | | |
| cos decay end ratio | 1 | | |
| total steps | 550550 | 550550 | 330330 |
| channel_mult | [1,1,2,2,4,8] | | |
| channel | 256 | | |
| num_res_blocks | 4 | | |
| generative decoder | False | True | True |
| per GPU batchsize | 6 | | |
| global batchsize | 1056 | | |
| GPU number | 176 H20 | | |

Table 36: **Large scale training on general-domain dataset at 384 compression ratio.**

| config | 384 compression ratio SOTA | | |
|---|---|---|---|
| training data | general domain dataset | | |
| image size | [256, 256] | | |
| data augmentation | random crop | | |
| downsample | $16 \times 16$ | | |
| ema | True | | |
| $g$ (group number) | 2 | | |
| $d'$ (group channel) | 8 | | |
| optimizer | Adam | | |
| optimizer momentum | $\beta_1, \beta_2$=0.5, 0.9 | | |
| weight decay | 0 | | |
| learning rate schedule | consistent | | |
| learning rate | 1e-4 | 1e-5 | 1e-6 |
| warmup steps | 0 | | |
| cos decay end ratio | 1 | | |
| total steps | 550550 | 200200 | 50050 |
| channel_mult | [1,1,2,2,4] | | |
| channel | 256 | | |
| num_res_blocks | 4 | | |
| generative decoder | False | False | False |
| per GPU batchsize | 8 | | |
| global batchsize | 1024 | | |
| GPU number | 128 H20 | | |

Table 37: **Large scale training on general-domain dataset at 192 compression ratio.**

| config | 192 compression ratio SOTA | | |
|---|---|---|---|
| training data | general domain dataset | | |
| image size | [256, 256] | | |
| data augmentation | random crop | | |
| downsample | $16 \times 16$ | | |
| ema | True | | |
| $g$ (group number) | 4 | | |
| $d'$ (group channel) | 8 | | |
| optimizer | Adam | | |
| optimizer momentum | $\beta_1, \beta_2$=0.5, 0.9 | | |
| weight decay | 0 | | |
| learning rate schedule | consistent | | |
| learning rate | 1e-4 | 1e-5 | 1e-6 |
| warmup steps | 0 | | |
| cos decay end ratio | 1 | | |
| total steps | 470470 | 90090 | 10010 |
| channel_mult | [1,1,2,2,4] | | |
| channel | 256 | | |
| num_res_blocks | 4 | | |
| generative decoder | False | False | False |
| per GPU batchsize | 8 | | |
| global batchsize | 1024 | | |
| GPU number | 128 H20 | | |

Table 38: **Large scale training on general-domain dataset at 48 compression ratio.**

| config | 48 compression ratio SOTA | | |
|---|---|---|---|
| training data | general domain dataset | | |
| image size | [256, 256] | | |
| data augmentation | random crop | | |
| downsample | $8 \times 8$ | | |
| ema | True | | |
| $g$ (group number) | 4 | | |
| $d'$ (group channel) | 8 | | |
| optimizer | Adam | | |
| optimizer momentum | $\beta_1, \beta_2$=0.5, 0.9 | | |
| weight decay | 0 | | |
| learning rate schedule | consistent | | |
| learning rate | 1e-4 | 1e-5 | 1e-6 |
| warmup steps | 0 | | |
| cos decay end ratio | 1 | | |
| total steps | 200200 | 200200 | 20020 |
| channel_mult | [1,2,2,4] | | |
| channel | 256 | | |
| num_res_blocks | 4 | | |
| generative decoder | False | | |
| per GPU batchsize | 8 | | |
| global batchsize | 1024 | | |
| GPU number | 128 H20 | | |

Table 39: **Large scale training on general-domain dataset at 24 compression ratio.**

| config | 24 compression ratio SOTA | | |
|---|---|---|---|
| training data | general domain dataset | | |
| image size | [256, 256] | | |
| data augmentation | random crop | | |
| downsample | $8 \times 8$ | | |
| ema | True | | |
| $g$ (group number) | 8 | | |
| $d'$ (group channel) | 8 | | |
| optimizer | Adam | | |
| optimizer momentum | $\beta_1, \beta_2$=0.5, 0.9 | | |
| weight decay | 0 | | |
| learning rate schedule | consistent | | |
| learning rate | 1e-4 | 1e-5 | 1e-5 |
| warmup steps | 0 | | |
| cos decay end ratio | 1 | | |
| total steps | 300300 | 50050 | 400400 |
| channel_mult | [1,2,2,4] | | |
| channel | 256 | | |
| num_res_blocks | 4 | | |
| generative decoder | False | | |
| per GPU batchsize | 8 | | |
| global batchsize | 1024 | 1024 | 4096 |
| GPU number | 128 H20 | 128 H20 | 512 H20 |

## F    DECLARATION OF USE OF LARGE LANGUAGE MODELS (LLM)

We affirm that this paper was primarily written by the authors. Large Language Models (LLMs) were utilized solely as general-purpose assistive tools for language refinement, grammar correction, and stylistic improvements during the writing process. Specifically, Gemini 2.5 Pro (DeepMind, 2025) was employed for minor text polishing and rephrasing to enhance clarity and readability. No LLM was used for conceptual ideation, experimental design, data analysis, or generating any substantive content of the research.

