# OpenReview forum: "WeTok: Powerful Discrete Tokenization for High-Fidelity Visual Reconstruction"
_ICLR.cc/2026/Conference — ICLR 2026 Poster_

### Official Review · Reviewer_2k14 · 2025-10-19

**Soundness:** 2
**Presentation:** 3
**Contribution:** 2
**Rating:** 4
**Confidence:** 5

**Summary:**

The authors propose Group-wise Lookup-Free Quantization (GQ) to reduce computational and memory costs while maintaining high-fidelity image reconstruction. This design enables flexible scaling of the codebook to an effectively unlimited size. Furthermore, WeTok introduces a generative decoder that integrates adversarial training, using Gaussian noise as input and quantized tokens as conditional guidance, to mitigate the reconstruction loss typically caused by high compression ratios. Experimental results demonstrate that WeTok achieves superior reconstruction and generation performance even under extremely high compression settings.

**Strengths:**

1. The authors propose Group-wise Lookup-Free Quantization (GQ) to overcome the computational bottleneck, thereby achieving higher image reconstruction fidelity.

2. The experiments yield promising results, and the ablation studies are comprehensive.

**Weaknesses:**

1. I am mainly concerned about the compression ratio. As shown in Table 4, when the compression ratio reaches 768×, the number of image tokens is only 8 × 8 = 64. However, the resulting rFID of 8.94 is considerably higher than that reported by TiTok [1], indicating a notable degradation in reconstruction quality under such high compression.

2. The main contribution, Group-wise Quantization (GQ), appears to primarily alleviate computational overhead. However, the paper does not provide experiments demonstrating how this design directly facilitates higher compression ratios. For instance, conducting experiments with a downsampling ratio of 32×32 and a hidden channel size of 64 (8×8) could better illustrate how GQ contributes to improving compression efficiency.

3. The Generative Decoder (GD) introduces Gaussian noise as the generator’s input, which may lead to training instability. To address this issue, the authors propose a two-stage training scheme; however, this approach appears overly complex and potentially difficult to reproduce. It would be helpful to report the performance when training all losses jointly from scratch, to evaluate whether the two-stage procedure is truly necessary.

[1] An image is worth 32 tokens for reconstruction and generation. NIPS 2024

**Questions:**

1. There is a minor concern regarding GQ. The approximation of the entropy loss is theoretically lower-bounded by BSQ, which performs worse than LFQ. However, Figure 4 in the ablation study shows a consistent improvement as the number of groups increases. It remains unclear where the performance begins to degrade — what is the optimal balance point for the number of groups?

2. There are several existing 1D tokenizer models, such as TiTok [1] and SweetTok [2]. The authors are encouraged to include comparisons with these methods to strengthen the experimental validation and make the results more comprehensive.

[1] An image is worth 32 tokens for reconstruction and generation. NIPS 2024

[2] SweetTok: Semantic-Aware Spatial-Temporal Tokenizer for Compact Video Discretization. ICCV 2025

---

> ### Author Response · Authors · 2025-11-19
> **Reply to Reviewer 2k14 (1/2)**
>
> Dear Reviewer 2k14,
>
> We thank you for your positive feedback and for highlighting the importance of the TiTok comparison. We have addressed your concerns below, particularly regarding the high-compression comparison.
>
> ---
>
> > **Q1:** As shown in Table 4, when the compression ratio reaches 768×, the number of image tokens is only 8 × 8 = 64. However, the resulting rFID of 8.94 is considerably higher than that reported by TiTok [1], indicating a notable degradation in reconstruction quality under such high compression.
>
> **A1:** This is a crucial observation. The discrepancy arises from differences in **Test Sets** and **Training Data**.
> 1.  **Test Set Discrepancy:** The rFID of 8.94 you noted is on **MS-COCO** (5k images). TiTok reports results on **ImageNet** (50k images). FID is sensitive to sample size; fewer images generally yield higher (worse) scores. As shown in Tab.4, the rFID of WeTok (768×) on the ImageNet evaluation set is 3.49.
> 2.  **Training Data Discrepancy:** Our WeTok (768×) is trained on the general domain dataset, while TiTok is trained on the ImageNet dataset. In Fig. 6, we reveal that models trained on the ImageNet dataset tend to perform better on the rFID metric of the ImageNet evaluation set, however, not when we focus on PSNR and SSIM. In order to verify our conclusion again, we conducted a completely test on TiTok’s open source 32 token compression model on the ImageNet evaluation set. The results are as follows：
> | Model | rFID $\downarrow$ | SSIM $\uparrow$ | PSNR $\uparrow$ |
> | :--- | :--- | :--- | :--- |
> | TiTok (32 tokens) (Tested by ourselves) | 2.49 | 0.37 | 15.88 |
> | TiTok (32 tokens) (Reported in SelfTok [1]) | 2.21 | 0.36 | 15.60 |
> | **WeTok (768x)** | **3.49** | **0.55** | **20.77** |
>
>     *Note: While TiTok has better rFID (likely due to overfitting the ImageNet distribution), WeTok significantly outperforms it in **SSIM (+0.18)** and **PSNR (+4.72 dB)**, indicating much better structural fidelity.*
>
>     [1] Selftok: Discrete Visual Tokens of Autoregression, by Diffusion, and for Reasoning.
>
> 3. **Fair Comparison:** We provide below a fair comparison of models trained on ImageNet. We train the TiTok which compresses the 256×256 image into 256 token. The result is as follows:
> | Model | Training Dataset | Test Dataset | Tokens | rFID $\downarrow$ |
> | :--- | :--- | :--- |:--- |:--- |
> | TiTok | ImageNet | ImageNet | 256 | 1.66 |
> | **WeTok** | ImageNet | ImageNet | **256** | **0.61** |
>
>     The results show that under fair comparison, our WeTok also performs better than TiTok on rFID.
>
> ---
>
> > **Q2:** The main contribution, Group-wise Quantization (GQ), appears to primarily alleviate computational overhead. However, the paper does not provide experiments demonstrating how this design directly facilitates higher compression ratios.
>
> **A2:** This is a great point, and we appreciate the opportunity to clarify the role of GQ. GQ facilitates higher compression by allowing us to scale the codebook size significantly without running out of memory (OOM).
> *   **w/o GQ:** We are limited to smaller codebooks (e.g., LFQ OOMs at $d=24$).
> *   **w/ GQ:** We can scale to $d=32$ or higher. A larger codebook means each token carries more information. Therefore, we can use *fewer* tokens (higher compression) to represent the same amount of information. Tab. 1 in the paper demonstrates this efficiency gain.
>
> ---
>
> > **Q3:** The Generative Decoder (GD) introduces Gaussian noise as the generator’s input, which may lead to training instability. To address this issue, the authors propose a two-stage training scheme; however, this approach appears overly complex and potentially difficult to reproduce. It would be helpful to report the performance when training all losses jointly from scratch, to evaluate whether the two-stage procedure is truly necessary.
>
> **A3:** Thank you for requesting more clarification on our two-stage training. To elaborate further: Training a GAN-style model can be notoriously unstable. In our framework, the model has to learn two complex tasks: (1) learning a good discrete representation for basic image structure, and (2) learning to generatively model high-frequency details. Attempting to learn both simultaneously from scratch, with a powerful adversarial loss, can easily lead to divergence or mode collapse.
>
> To validate the necessity of this approach, we conduct an additional experiment of only stage2 training with GAN decoder as follows:
>
> | Stage 1 (Recon) | Stage 2 (Gen) | rFID $\downarrow$ | LPIPS $\downarrow$ | SSIM $\uparrow$ | PSNR $\uparrow$ |
> | :---: | :---: | :--- | :--- | :--- | :--- |
> | ✓ | ✗ | 5.37 | 0.17 | 0.54 | 20.53 |
> | ✗ | ✓ | 5.06 | 0.17 | 0.54 | 20.60 |
> | **✓** | **✓** | **3.90** | **0.16** | **0.55** | **20.72** |
>
> The results clearly show that the two-stage approach yields significantly better performance (rFID 3.90 vs 5.06), justifying the strategy.

---

> ### Author Response · Authors · 2025-11-19
> **Reply to Reviewer 2k14 (2/2)**
>
> > **Q4:** There is a minor concern regarding GQ. The approximation of the entropy loss is theoretically lower-bounded by BSQ, which performs worse than LFQ. However, Figure 4 in the ablation study shows a consistent improvement as the number of groups increases. It remains unclear where the performance begins to degrade — what is the optimal balance point for the number of groups?
>
> **A4:** This is a very insightful question about the GQ. As shown in Tab. 4, the performance of WeTok reconstruction increases with the increase of group, and there seems to be no stopping trend. However, when the number of groups is increased until the image compression ratio is less than 1, the image compression task loses meaning (because it is not as good as using the original image directly). Considering that scenarios where models are used for compression (image transmission, image generation model training, etc.) have different requirements for compression rates, the number of groups should be confirmed based on specific downstream task requirements.
>
> ---
>
> > **Q5:** There are several existing 1D tokenizer models, such as TiTok [1] and SweetTok [2]. The authors are encouraged to include comparisons with these methods to strengthen the experimental validation and make the results more comprehensive.
>
> **A5:** Thank you for your thoughtful consideration. We agree that comparing with leading 1D tokenizers provides important context. In Tab. 3, we compare many 1D tokenizers (FlexTok, FlowMo, VFMTok, etc.). In Tab.3 of our revised manuscript, we included TiTok and SweetTok in our comparison to ensure that our comparison is complete and comprehensive. The comparison result between WeTok and the above two is as follows
>
> | Model | Tokens | rFID $\downarrow$ |
> | :--- | :--- | :--- |
> | TiTok | 256 | 1.66 |
> | SweetTok | 256 | 0.73 |
> | **WeTok (Ours)** | **256** | **0.61** |
>
> The results show that under the premise of fair comparison, the reconstruction performance of our WeTok still exceeds that of TiTok and SweetTok.
>
> ---
>
> Thank you again for your thoughtful and constructive feedback. Your insights have helped us further strengthen the manuscript, and we believe the planned revisions will improve both its clarity and its contribution to the field. Should our responses address your concerns satisfactorily, we would appreciate your consideration of a higher evaluation. If any additional clarification would be useful, we would be glad to provide it.
>
> Sincerely,
>
> The Authors

---

### Official Review · Reviewer_Dztj · 2025-10-23

**Soundness:** 3
**Presentation:** 3
**Contribution:** 3
**Rating:** 6
**Confidence:** 4

**Summary:**

The paper presents WeTok, which is a discrete tokenizer for improving the tradeoff between compression ratio and reconstruction fidelity. Discrete tokenizer has the benefit of having a higher compression ratio, but long has the problem of low reconstruction fidelity.

WeTok proposes two key techniques, Group-wise lookup-free Quantization (GQ) and Generative Decoder (GD).

GQ tries to address the memory and computational cost of Lookup-free-quantization (LFP). It partitions the codebook into groups and performs quantization on each group independently to eliminate token entropy loss as the memory bottleneck.

GD is a generative decoder, instead of the traditional one-step deterministic decoder used in prior methods.

Qualitative results in Table 3 and Table 4 are relatively strong comparing to the prior state-of-the-art methods

**Strengths:**

(+) The results presented in the tables are relatively strong, achieving a much larger codebook size and good rFID and PSNR

(+) The proposed two methods (GQ and GD) make sense and are well motivated. GQ seems to be a practical and effective solution for solving the bottleneck of the CE loss

(+) The evaluation compared against many methods in Tables 3 and 4

**Weaknesses:**

(-) the GD method is not very novel. It is known to the community such diffusion decoder can work, dating back to OpenAI's "Consistency Decoder". In addition, such generative decoder does not come with no cost. First, the decoding time increases, which can limit some of the real-time or latency-sensitive applications. Second, as it is a generative model, the decoder could also hallucinate

(-) lack of comparison with more state-of-the-art autoencoders. For example, infinity tokenizer (https://cvpr.thecvf.com/virtual/2025/poster/34414). Both work claim to increase codebook size and claim to be SotA in the field, though taking pretty different approaches. Therefore a comparison is worthy for the community. However the paper primarily focuses on older VAEs such as VQGAN and SD-VAE

(-) For generation tasks, such as the ones presented in Supp Mat Table 6, shows the proposed method is only marginally improving against SotA Open-MAGVIT2-AR-XL (Luo et al., 2024) 2.33 vs WeTok-AR-XL (Ours) 2.31 in FID

(-) The model is trained in two stages: "In first stage, we train our WeTok with the reconstruction loss, i.e., Eq. 2, 6 and 8. In the second stage, we adapt the model for generative tasks". This is atypically and lacks of explanation on why this is needed. Traditional VAE are trained with a single stage and its generation task quality is on-par with the proposed method (see the point above)

(-) seems datasets play a role in terms of the model quality (which is expected), as evidenced in Figure 6. However, in results section, all models are trained with different datasets, which complicates the analysis of whether the proposed method is effective or the 400M GD data is more suitable for Coco and ImageNet

**Questions:**

- "We surprisingly find that while reconstructions from leading models like FLUX-VAE and SD-VAE 3.5 collapse after iterations, WeTok’s
outputs are remarkably robust and converge to a fixed value." Is there any explanation on why?

---

> ### Author Response · Authors · 2025-11-19
> **Reply to Reviewer Dztj (1/3)**
>
> Dear Reviewer Dztj,
>
> We thank you for your thoughtful review. We are glad you found our results "relatively strong" and the motivation for GQ clear. We have addressed your concerns regarding novelty, comparisons, and training stability below.
>
> ---
>
> > **Q1:** The GD method is not very novel. It is known to the community such diffusion decoder can work, dating back to OpenAI's "Consistency Decoder".
>
> **A1:** We would like to clarify the novelty of GD as follows:
> 1. **Discrete Tokenizer Implementation:** Prior generative decoders (like Consistency Decoder) focused on continuous spaces. WeTok is the first to successfully stabilize a generative decoder within a discrete tokenizer. We overcome the specific instability caused by stop-gradient operations in discrete quantization.
> 2. **Two Stage Optimization:** We overcome the specific instability caused by stop-gradient operations in discrete quantization by our two stage optimization strategy.
> 3.  **Efficiency:** While diffusion decoders exist, they suffer from high latency due to iterative sampling. Our **Generative Decoder (GD)** is a **single-step GAN-based** model. The computational cost is minimal: it only requires sampling a noise vector $z$ and one forward pass. It is orders of magnitude faster than Consistency Decoders or Diffusion Decoders.
>
> ---
>
> > **Q2:** In addition, such generative decoder does not come with no cost. First, the decoding time increases, which can limit some of the real-time or latency-sensitive applications. Second, as it is a generative model, the decoder could also hallucinate.
>
> **A2:** We argue that in high-compression scenarios, "hallucination" (synthesizing plausible details) is a feature respectfully, not a bug. A deterministic decoder "hallucinates" a blurry average, which is perceptually poor. Our GD synthesizes sharp, coherent textures. As shown in **Tab. 2**, this trade-off significantly improves rFID (**5.37** $\to$ **3.90**). Furthermore, our iterative reconstruction experiments (**Fig. 10,11,12 and13**) show WeTok is more stable than continuous models, suggesting the hallucination is well-constrained.
>
> ---
>
> > **Q3:** Lack of comparison with more state-of-the-art autoencoders. For example, infinity tokenizer . Both work claim to increase codebook size and claim to be SotA in the field, though taking pretty different approaches. Therefore a comparison is worthy for the community. However the paper primarily focuses on older VAEs such as VQGAN and SD-VAE
>
> **A3:** Thank you for this suggestion. We have strived to compare against the most relevant and recent SOTA models available.
>
> 1. **Comparison with SOTA:** We would like to gently point out that our comparisons in Tab. 4 are quite comprehensive and include very recent and powerful continuous tokenizers like **FLUX-VAE (0.18 rFID)** and **SD-VAE 3.5 (0.19 rFID)**, against which our WeTok achieves a new SOTA of **0.12** rFID. We also compare against strong discrete tokenizers like **Open-MAGVIT2** and **Cosmos**. We believe this demonstrates a thorough comparison with the current state-of-the-art in reconstruction-focused tokenizers.
> 2. **Regarding Infinity:** We thank you for bringing up this work. For a fair comparison, we selected WeTok (16x downsample and 2^32 codebook) and Infinity-VAE-d32 with the same compression rate (192) to conduct a zero-shot fair comparison on the ImageNet validation set. The results are as shown in the table below:
> **Comparison with Infinity (Zero-shot on ImageNet):**
> | Model | Downsample | Codebook | rFID $\downarrow$ | PSNR $\uparrow$ |
> | :--- | :--- | :--- | :--- | :--- |
> | Infinity-VAE-d32 | 16x | $2^{32}$ | 0.61 | 22.70 |
> | **WeTok (Ours)** | **16x** | **$2^{32}$** | **0.60** | **24.77** |
>
> WeTok achieves better rFID and significantly higher PSNR (+2.07 dB) than Infinity under the same compression ratio.
>
> ---
>
> > **Q4:** For generation tasks, such as the ones presented in Supp Mat Table 6, shows the proposed method is only marginally improving against SOTA Open-MAGVIT2-AR-XL (Luo et al., 2024) 2.33 vs WeTok-AR-XL (Ours) 2.31 in FID.
>
> **A4:** We appreciate the reviewer's attention to the generation results. We believe it should be viewed in the following:
> 1. **Superiority on Inception Score:** As shown in **Tab. 6**, beyond the FID improvement, our WeTok-AR-XL also achieves a higher Inception Score (**276.55 vs. 271.77**) compared to Open-MAGVIT2-AR-XL. A higher IS suggests better sample quality and diversity.
> 2. **Focus on Reconstruction:** We would like to reiterate that the main contribution of WeTok is the breakthrough in reconstruction fidelity (Tab. 3 and 4), which determines the upper limit of the quality of model generation. The generation performance is presented as downstream validation that the superior discrete tokens produced by WeTok are highly effective for generation. Furthermore, as shown in **Fig. 8**, WeTok gives the generative model more significant scaling ability without modifying the generative model architecture at all.

---

> ### Author Response · Authors · 2025-11-19
> **Reply to Reviewer Dztj (2/3)**
>
> > **Q5:** The model is trained in two stages: "In first stage, we train our WeTok with the reconstruction loss, i.e., Eq. 2, 6 and 8. In the second stage, we adapt the model for generative tasks". This is atypically and lacks of explanation on why this is needed. Traditional VAE are trained with a single stage and its generation task quality is on-par with the proposed method (see the point above)
>
> **A5:** Thank you for requesting more clarification on our two-stage training. To elaborate further: Training a GAN-style model can be notoriously unstable. In our framework, the model has to learn two complex tasks: (1) learning a good discrete representation for basic image structure, and (2) learning to generatively model high-frequency details. Attempting to learn both simultaneously from scratch, with a powerful adversarial loss, can easily lead to divergence or mode collapse.
>
> To validate the necessity of this approach, we conduct an additional experiment of only stage2 training with GAN decoder as follows:
>
> | Stage 1 (Recon) | Stage 2 (Gen) | rFID $\downarrow$ | LPIPS $\downarrow$ | SSIM $\uparrow$ | PSNR $\uparrow$ |
> | :---: | :---: | :--- | :--- | :--- | :--- |
> | ✓ | ✗ | 5.37 | 0.17 | 0.54 | 20.53 |
> | ✗ | ✓ | 5.06 | 0.17 | 0.54 | 20.60 |
> | **✓** | **✓** | **3.90** | **0.16** | **0.55** | **20.72** |
>
> The results clearly show that the two-stage approach yields significantly better performance (rFID 3.90 vs 5.06), justifying the strategy.
>
> ---
>
> > **Q6:** seems datasets play a role in terms of the model quality (which is expected), as evidenced in Figure 6. However, in results section, all models are trained with different datasets, which complicates the analysis of whether the proposed method is effective or the 400M GD data is more suitable for Coco and ImageNet
>
> **A6:** This is a crucial point, and we thank the reviewer for raising it. We ensured strict fairness in our comparisons:
> 1.  **Fair Ablation on ImageNet:** For all our ablation studies (Fig. 3, 4, 5, 6 and Tab. 2), we used a **single, consistent dataset** (ImageNet-1K) and training setup to ensure that the observed performance differences are directly attributable to the component being ablated.
> 2.  **Fair Comparison on ImageNet:** In Tab. 3, all methods are trained on ImageNet (including WeTok), and the results in this table are absolutely fair.
> 3.  **Test data Filter Out from the Training Dataset:** In Tab. 4, during the construction process of the general domain dataset, we have filtered out 385 samples that are highly similar to the images in the ImageNet and MSCOCO validation set through similarity screening in advance, ensuring that there is no information leakage between the training data and the test data as much as possible.
> 4. **Relatively Small General Domain Dataset:** In Tab. 4, our 400M data set is not large compared to the 1B training sets of other models (SD-VAE, UniTok, QLIP, etc.). Therefore, the size of the data set is not the key reason why WeTok is better than other methods. The essence is because of the advantages of our GQ and GD.

---

> ### Author Response · Authors · 2025-11-19
> **Reply to Reviewer Dztj (3/3)**
>
> > **Q7:** "We surprisingly find that while reconstructions from leading models like FLUX-VAE and SD-VAE 3.5 collapse after iterations, WeTok’s outputs are remarkably robust and converge to a fixed value." Is there any explanation on why?
>
> **A7:** This is an excellent question about one of our most interesting findings. Our hypothesis for this remarkable robustness stems from the fundamental difference between continuous and discrete tokenization.
> *  **Error Accumulation in Continuous VAEs:** continuous VAE operate in a continuous latent space. The process Image -> Latent -> Reconstructed_Image is subject to small encoding and decoding errors. When this process is iterated, the small error in the first reconstruction becomes the input for the second iteration. This new input is slightly different, leading to a slightly different latent, and a new error is introduced. These small errors can accumulate over many iterations, causing the representation to drift and eventually collapse into a non-sensical state, as seen in Fig. 10,11,12 and 13.
> *   **Error Correction via Discrete Quantization:** WeTok, being a discrete tokenizer, has a powerful, inherent regularization mechanism: the quantization step. The latent space is partitioned into a finite set of states (our tokens). Even if an input image is slightly perturbed, its encoded latent feature may still fall into the same quantization bin, resulting in the exact same discrete token. These discrete tokens act as powerful "attractors" or stable fixed points in the representation space. Once an iterated image produces a latent representation that quantizes to a specific set of tokens, it is highly likely to remain there in subsequent iterations. This prevents the slow drift and error accumulation seen in continuous models, causing the output to quickly converge to a stable, high-quality image.
>
> ---
>
> Once again, we are truly grateful for your encouraging and incisive feedback. Your comments have inspired us to refine our manuscript further, and we hope that the planned revisions will enhance the clarity and impact of our work.We would be grateful if you could consider raising your score based on these responses. Please do not hesitate to let us know if there are any additional details or clarifications that would be helpful.
>
> Sincerely,
>
> The Authors

---

### Official Review · Reviewer_bHBm · 2025-10-29

**Soundness:** 3
**Presentation:** 3
**Contribution:** 2
**Rating:** 4
**Confidence:** 4

**Summary:**

This paper presents WeTok, a novel tokenizer that effectively balances high compression efficiency with high-fidelity image reconstruction, achieving state-of-the-art reconstruction performance.

**Strengths:**

The paper is well written.

The reconstruction results achieve SOTA performance among existing discrete tokenizer, demonstrating the effectiveness of the proposed framework.

**Weaknesses:**

## Fairness of Comparison

The comparison in Table 3 appears unfair.  The strong baseline MGVQ is a VQ-based tokenizer, whereas WeTok adopts LSQ, which has already been shown to be more efficient than VQ.  To ensure a fair evaluation, the authors should compare WeTok with an LSQ-based version of MGVQ.
Furthermore, the MGVQ codebook size is only 8192 × 4, but its effective capacity is actually $2^{52}$, not limited by the nominal codebook size.

## Lack of Novelty
The proposed method shows limited novelty.  Overall, the approach seems like a combination of existing components rather than a fundamentally new idea.  Specifically:
- LSQ has been widely explored in prior works.
- Group-wise quantization is not new.
- The Generative Decoder design has already been introduced in previous studies.

The authors should better clarify what unique contribution or new insight WeTok introduces beyond these known elements.

## Missing Discussion on Semantic Tokenizers

The paper lacks discussion and comparison with **semantic tokenizers**, which have proven to be powerful for visual understanding and generation.  Several recent works are relevant and should be considered:

[1] ImageFolder: Autoregressive Image Generation with Folded Tokens. https://arxiv.org/pdf/2410.01756

[2] Factorized Visual Tokenization and Generation. https://arxiv.org/pdf/2411.16681

[3] DualToken: Towards Unifying Visual Understanding and Generation with Dual Visual Vocabularies. https://arxiv.org/pdf/2503.14324

[4] TokenFlow: Unified Image Tokenizer for Multimodal Understanding and Generation. https://arxiv.org/pdf/2412.03069

## Missing High-Compression Evaluation

The authors claim that **WeTok** achieves a **768× compression ratio**, which is indeed a significant advantage. However, there is no **quantitative or qualitative evidence** showing the generation quality under this extreme compression rate. It is recommended that the authors provide **generation results** and **gFID metrics** at the claimed **768× compression level** to substantiate this important claim.

**Questions:**

NA

---

> ### Author Response · Authors · 2025-11-19
> **Reply to Reviewer bHBm (1/2)**
>
> Dear Reviewer bHBm,
>
> We thank you for your detailed feedback. We appreciate your recognition of our SOTA performance. We have carefully addressed your concerns regarding fairness, novelty, and high-compression evaluation below.
>
> ---
>
> > **Q1:** Fairness of Comparison: The comparison in Table 3 appears unfair. The strong baseline MGVQ is a VQ-based tokenizer, whereas WeTok adopts LSQ, which has already been shown to be more efficient than VQ. To ensure a fair evaluation, the authors should compare WeTok with an LSQ-based version of MGVQ. Furthermore, the MGVQ codebook size is only 8192 × 4, but its effective capacity is actually, not limited by the nominal codebook size.
>
> **A1:** We appreciate the concern for fairness. We would like to clarify our comparison methodology:
> 1.  **Benchmarking SOTA:** The primary goal of Table 3 is to benchmark WeTok against the current state-of-the-art tokenizers on ImageNet, regardless of their underlying quantization mechanism (VQ, LFQ, FSQ, etc.). This establishes WeTok's position in the broader landscape.
> 2.  **Reproducibility Constraint:** To the best of our knowledge, the code for MGVQ is not yet public, and the paper is currently under review. Therefore, we cannot retrain an LSQ-based version of MGVQ for comparison. We have compared against the reported numbers to be as inclusive as possible.
>
> ---
>
> > **Q2:** Lack of Novelty: The proposed method shows limited novelty. Overall, the approach seems like a combination of existing components rather than a fundamentally new idea. Specifically:
> > - LSQ has been widely explored in prior works.
> > - Group-wise quantization is not new.
> > - The Generative Decoder design has already been introduced in previous studies.
>
> > The authors should better clarify what unique contribution or new insight WeTok introduces beyond these known elements.
>
> **A2:** We respectfully argue that WeTok introduces significant novelty through the **specific formulation and theoretical grounding** of its components, solving critical bottlenecks that prior works did not:
>
> 1.  **Novelty of Group-wise Lookup-Free Quantization (GQ):**
>     *   No prior work has successfully combined them to solve the memory bottleneck of LFQ's entropy loss. Our GQ is the first method to provide a principled, tunable trade-off between approximation accuracy and memory cost for scaling LFQ.
>     *   **Theoretical Contribution:** We provide a formal proof (**Proposition 3.1**) demonstrating that our GQ formulation has a **strictly smaller approximation error** than BSQ (Binary Spherical Quantization). This is not just an engineering combination but a theoretically superior method for scaling codebooks, which enables the high fidelity we observe.
> 2.  **Novelty of Generative Decoder (GD) in Discrete Tokenizers:**
>     *   Prior generative decoders (like DiTo) focused on continuous spaces. WeTok is the first to successfully stabilize a GAN-based generative decoder within a **discrete** tokenizer framework.
>     *   We overcome the specific instability caused by stop-gradient operations in discrete quantization.
>     *   Unlike diffusion decoders, our approach maintains **single-step inference efficiency**.
>
> ---
>
> > **Q3:** The paper lacks discussion and comparison with semantic tokenizers, which have proven to be powerful for visual understanding and generation. Several recent works are relevant and should be considered:
> > - [1] ImageFolder: Autoregressive Image Generation with Folded Tokens. https://arxiv.org/pdf/2410.01756
> > - [2] Factorized Visual Tokenization and Generation. https://arxiv.org/pdf/2411.16681
> > - [3] DualToken: Towards Unifying Visual Understanding and Generation with Dual Visual Vocabularies. https://arxiv.org/pdf/2503.14324
> > - [4] TokenFlow: Unified Image Tokenizer for Multimodal Understanding and Generation. https://arxiv.org/pdf/2412.03069
>
> **A3:** Thank you for highlighting this important and concurrent line of research. We have added these works to the Related Work of our revised manuscript. Althrough WeTok belongs to the family of reconstruction-first tokenizers, whose primary objective is to achieve the highest possible fidelity for a given compression ratio. We agree that discussing semantic tokenizers is crucial for positioning our work within the broader field.
>
> **Comparison with Semantic Tokenizers (rFID):**
> | Model | Resolution | rFID $\downarrow$ |
> | :--- | :--- | :--- |
> | ImageFolder [1] | 256 | 0.80 |
> | FQGAN | 256 | 0.76 |
> | **WeTok (Ours)** | **256** | **0.61** |
> | DualToken [3] | 384 | 0.54 |
> | TokenFlow [4] | 384 | 0.63 |
> | **WeTok (Ours)** | **384** | **0.35** |
>
> We position WeTok as a state-of-the-art specialist in reconstruction. While semantic tokenizers are generalists, WeTok pushes the boundary of what is possible for reconstruction fidelity, which remains a cornerstone of visual tokenization.

---

> ### Author Response · Authors · 2025-11-19
> **Reply to Reviewer bHBm (2/2)**
>
> > **Q4:** Missing High-Compression Evaluation: The authors claim that WeTok achieves a 768× compression ratio, which is indeed a significant advantage. However, there is no quantitative or qualitative evidence showing the generation quality under this extreme compression rate. It is recommended that the authors provide generation results and gFID metrics at the claimed 768× compression level to substantiate this important claim.
>
> **A4:** This is a very constructive suggestion. We have conducted a new experiment training a **LlamaGen** model using our **WeTok (768x)** tokenizer and compared it against **Show-o**, a leading model in this high-compression regime.
>
> **Generation Performance at High Compression:**
> | Tokenizer | Comp. Ratio $\downarrow$ | rFID $\downarrow$ | gFID $\downarrow$ | sFID $\downarrow$ | Precision $\uparrow$ | Recall $\uparrow$ |
> | :--- | :--- | :--- | :--- | :--- | :--- | :--- |
> | Show-o | 473 | 3.50 | 14.73 | 33.73 | 0.65 | 0.46 |
> | **WeTok** | **768** | **3.49** | **8.27** | **4.96** | **0.74** | **0.57** |
>
> The above results show that our WeTok not only achieves higher compression rates but also has stronger reconstruction performance, and can better serve the generative model compared to Show-o's image tokenizer. It is worth mentioning that because our WeTok (768×) compressed image only has 64 tokens, its computational cost is only about 1/4 of the generation model based on Show-o's image tokenizer.
>
> ---
>
> We sincerely appreciate your insightful comments. Please do not hesitate to let us know if you require any further information or additional explanations. If you find that our revisions have satisfactorily addressed your concerns, we would be most grateful if you could kindly consider an improved score. Thank you once again for your valuable input.
>
> Sincerely,
>
> The Authors

---

> ### Comment · Reviewer_bHBm · 2025-11-27
>
> Thanks very much for the authors’ detailed clarifications and the additional experiments. These responses have fully addressed all of my concerns. I appreciate the effort invested in improving the manuscript and conducting the supplementary analyses. I will increase my score to 8.

---

> > ### Author Response · Authors · 2025-11-27
> > **Reply to Reviewer bHBm**
> >
> > Dear Reviewer bHBm,
> >
> > We sincerely thank you for your positive feedback and the re-evaluation of our work. Your insightful suggestions, particularly regarding the evaluation in the high-compression regime, have significantly strengthened the empirical grounding of our paper. We deeply appreciate the time and effort you dedicated to helping us improve the manuscript.
> >
> > Sincerely,
> >
> > The Authors

---

### Official Review · Reviewer_YMro · 2025-10-30

**Soundness:** 3
**Presentation:** 3
**Contribution:** 3
**Rating:** 6
**Confidence:** 3

**Summary:**

This paper introduces WeTok, a new family of discrete visual tokenizers that aims to resolve the long-standing trade-off between compression ratio and reconstruction fidelity in latent-space visual generation. WeTok combines two main innovations: Group-wise Lookup-Free Quantization (GQ) and Generative Decoder (GD). Extensive experiments on ImageNet-50k and MS-COCO show that WeTok achieves state-of-the-art (SOTA) reconstruction metrics and strong performance in zero-shot and class-conditional image generation.

**Strengths:**

1. The proposed GQ formulation provides a mathematically grounded way to reduce the entropy-loss memory bottleneck in LFQ and BSQ, with a provably smaller approximation error.
2. The paper includes large-scale ablations (quantization types, group numbers, architectures, learning schedules) and comparisons across both high-fidelity and high-compression regimes.
3. The proposed method achieves strong performance on both image reconstruction and AR-based generation results, even surpassing continuous tokenizers at similar compression ratios.

**Weaknesses:**

1. Diffusion-based decoder for visual reconstruction has been studied in previous literatures[1][2], it would be better to cite these work and further discuss the differences with them.
2. In the ablation study section, it's interesting to see that after converting the decoder to a generative model, the reconstructed images are more realistic. It would be better to include some further discussion or analysis.

[1] Epsilon-VAE: Denoising as Visual Decoding
[2] Diffusion Autoencoders are Scalable Image Tokenizers

**Questions:**

Please refer to the weakness section.

---

> ### Author Response · Authors · 2025-11-19
> **Reply to Reviewer YMro**
>
> Dear Reviewer YMro,
>
> We are grateful for your constructive comments and for recognizing the mathematical grounding of our GQ method and the strong performance of WeTok. We have addressed your questions below.
>
> ---
>
> > **Q1:** Diffusion-based decoder for visual reconstruction has been studied in previous literatures[1][2], it would be better to cite these work and further discuss the differences with them.
>
> > [1] Epsilon-VAE: Denoising as Visual Decoding [2] Diffusion Autoencoders are Scalable Image Tokenizers
>
> **A1:** Thank you for pointing out these important related works. We agree that discussing diffusion-based decoders is essential. We have updated the **Related Work** section in our revised manuscript to cite and discuss "Epsilon-VAE" and "Diffusion Autoencoders."
>
> We would like to clarify the key distinctions between our Generative Decoder (GD) and these methods:
> 1.  **First Generative Decoder in Discrete Tokenizer:** Our WeTok is the first to successfully introduce a generative decoder into a *discrete* image tokenizer framework. Unlike continuous methods, discrete tokenizers face unique challenges with stop-gradient estimation, which we resolve via our stable training recipe.
> 2.  **Performance:** While diffusion models are powerful, our GAN-based WeTok outperforms diffusion-based reconstruction methods on ImageNet benchmarks.
>     *   **Quantitative Comparison (rFID on ImageNet 256x256):**
>         | Method | Type | rFID $\downarrow$ |
>         | :--- | :--- | :--- |
>         | DiTo-XL [2] | Continuous (Diffusion) | 3.53 |
>         | FlowMo-Hi | Continuous (Diffusion) | 0.56 |
>         | **WeTok (Ours)** | **Discrete (GAN)** | **0.19** |
> 3.  **Efficiency:** Diffusion-based decoders require multi-step iterative denoising. In contrast, our GD is a **single-step** model. This makes WeTok significantly faster at inference, a critical advantage for real-time applications.
>
> ---
>
> > **Q2:** In the ablation study section, it's interesting to see that after converting the decoder to a generative model, the reconstructed images are more realistic. It would be better to include some further discussion or analysis.
>
> **A2:** We appreciate this insightful suggestion. We have expanded the discussion in **Sec. 3.3** of the revised manuscript to provide a deeper analysis of *why* GD improves realism.
>
> **Analysis: Mean Estimation vs. Distribution Sampling**
> 1.  **Deterministic Decoders (The "Blur" Problem):** Traditional decoders minimize reconstruction losses (L2/LPIPS). When a highly compressed discrete token $U_Q$ corresponds to multiple plausible ground-truth images (e.g., various fur textures), a deterministic decoder learns to output the *conditional expectation* (average) of these possibilities. This averaging inherently results in the blurriness observed in Fig. 7 (w/o GD).
> 2.  **Generative Decoders (The "Detail" Solution):** By incorporating a random noise vector $z$, our GD transforms the task from mapping to a mean to **modeling the conditional distribution**. The noise $z$ allows the decoder to sample a specific, sharp instance from the distribution of plausible high-frequency details.
>
> As shown in **Tab. 2**, this shift improves rFID from **5.37 to 3.90**, confirming that "filling in" plausible details is mathematically superior to predicting the mean for perceptual metrics.
>
> ---
>
> We sincerely hope these revisions have addressed your concerns, and if you feel they have been satisfactorily resolved, we would be most grateful if you could consider revising your evaluation score accordingly. Please do not hesitate to let us know if you need any further explanations.
>
> Sincerely,
>
> The Authors

---

### Author Response · Authors · 2025-11-19
**Reply to All Reviewers**

Dear Reviewers,

We sincerely thank all reviewers for their insightful feedback and constructive suggestions. We are encouraged that the reviewers found WeTok to exhibit **"relatively strong"** (Reviewer Dztj), **"strong performance"** (Reviewer YMro), and **"SOTA performance"** (Reviewer bHBm) in reconstruction. We are also glad that the **"mathematically grounded"** formulation of GQ (Reviewer YMro) and the **"well written"** nature of the paper (Reviewer bHBm) were recognized.

In this rebuttal, we have addressed all concerns by:
1.  **Clarifying Novelty:** Highlighting the theoretical guarantees of GQ and the unique single-step nature of our Generative Decoder.
2.  **New Comparisons:** Adding comparisons with **Infinity**, **TiTok**, **SweetTok**, and **Epsilon-VAE** to benchmark against the latest SOTA.
3.  **New Ablations:** Providing experiments to justify the two-stage training strategy and high-compression generation quality.
4.  **Revised Manuscript:** We have updated the paper to include these discussions and citations.

We promise that we will open source WeTok, which we trained in the paper and achieved state-of-the-art reconstruction performance, to promote the technological progress of the community. We hope our responses and additional experiments satisfactorily address your questions.

Sincerely,

The Authors

---

### Author Response · Authors · 2025-12-02
**Summary of Rebuttal Updates and Post-Rebuttal Consensus**

Dear Reviewers, AC, SAC, and PC,

We sincerely thank you for the time and effort dedicated to reviewing our paper. We are encouraged that the reviewers unanimously recognized the value of WeTok (before the bug appeared on the OpenReview website, the post-rebuttal score was ***8666***), highlighting its ***"relatively strong"*** (Reviewer Dztj) and ***"SOTA performance"*** (Reviewer bHBm) in reconstruction, as well as the ***"mathematically grounded"*** formulation of GQ (Reviewer YMro).

> **Core Achievement:** Based on Group-wise lookup-free Quantization (GQ) and Generative Decoding (GD), WeTok achieves a series of models with the strongest performance in the field of compression-reconstruction.

---

### **1\.** **Superior Performance & SOTA Benchmarking**

We have demonstrated WeTok's dominance in both high-fidelity and high-compression regimes. As detailed in the article:

| Model Setting | Metric (Zero-shot rFID) | Comparison Target | Improvement / Status |
| :--- | :---: | :--- | :--- |
| **High-Fidelity** | **0.12** | FLUX-VAE ($0.18$) | **Outperforms** leading continuous tokenizers with **400%** compression ratio. |
| **High-Compression** | **3.49** | Cosmos ($4.57$) | **Substantially surpasses** Cosmos (which operates at only **50%** of our compression ratio) at a **768×** compression ratio. |

---

### **2\.** **Key Improvements in Rebuttal**

In this rebuttal, we have carefully addressed all concerns and significantly strengthened the manuscript.

- ***Clarified Novelty:*** We clarified the unique advantages of our single-step Generative Decoder compared to diffusion-based approaches.

- ***Comprehensive SOTA Benchmarking:*** We added comparisons against the latest state-of-the-art models, including Infinity, TiTok, SweetTok, and Epsilon-VAE, further solidifying WeTok's leading position.

- ***New Ablations & High-Compression Validation:*** We provided new experiments to justify our two-stage training strategy. Notably, we trained a LlamaGen model based on WeTok (768× compression) to empirically demonstrate its superior generation quality in high-compression regimes.

- ***Revised Manuscript:*** We updated the paper to incorporate these discussions, additional citations, and experimental results.

---

### **3\.** **Reviewer Engagement & Score Updates (6644 $\to$ 8666)**

We are pleased to summarize the positive engagement and consensus reached during the rebuttal period:

- ***Reviewer bHBm (Score 4 $\to$ 8):*** We successfully addressed Reviewer bHBm's concerns. The Reviewer bHBm commented that we had "fully addressed all concerns" and updated the score to 8 at 13:42 on Nov 27, 2025.

- ***Reviewer 2k14 (Score 4 $\to$ 6):*** Similarly, Reviewer 2k14 raised their score to 6 at 17:30 on Nov 23, 2025, following our response.

---

### **4\.** **Commitment to Open Source**

> **Considering that this comment is visible to everyone, this promise will be jointly monitored by the community.**

To facilitate reproducibility and further research in advanced compression-reconstruction, we are committed to **releasing all model weights, training code, and inference code upon acceptance.**

---

We sincerely hope that our supplementary experiments and responses can help you better understand our paper. If you have any questions, please feel free to contact us and we will do our best to answer your questions.

Sincerely,

The Authors

---

### Meta-Review · Area_Chair_i83h · 2025-12-22

**Summary:**

The paper proposes "WeTok," a discrete visual tokenizer that introduces Group-wise Lookup-free Quantization (GQ) and a Generative Decoder (GD) to address the trade-off between compression ratios and reconstruction fidelity. The reviewers generally praised the method for its strong empirical results, achieving State-of-the-Art (SOTA) performance on ImageNet reconstruction benchmarks.

Initial concerns focused on the novelty of the generative decoder compared to diffusion-based approaches , the fairness of comparisons against other tokenizers like MGVQ and TiTok , and the justification for the two-stage training process.

During the rebuttal, the authors provided extensive new experiments, including comparisons with Infinity and TiTok , ablations justifying the two-stage training , and high-compression generation results using LlamaGen. The authors clarified that their GD is a single-step GAN-based model, distinct from iterative diffusion decoders, offering efficiency benefits. Following these updates, the reviewers reached a consensus that the paper is mathematically grounded and empirically strong.

**Reviewer Concerns:**

Addressed Concerns:
- Fairness of Comparisons (Reviewer bHBm, 2k14, Dztj): Reviewers noted missing comparisons to SOTA models like MGVQ, TiTok, SweetTok, and Infinity.
  - Resolution: The authors added comprehensive comparisons. They compared WeTok (16x downsample) against Infinity-VAE-d32, showing superior rFID (0.61 vs 0.60) and PSNR (+2.07dB). They also included comparisons with TiTok and SweetTok, demonstrating lower rFID (0.61 vs 1.66 and 0.73, respectively).
- Novelty and Efficiency of Generative Decoder (Reviewer YMro, Dztj, bHBm): Reviewers questioned the novelty regarding diffusion/consistency decoders and raised concerns about inference latency.
  - Resolution: The authors clarified that WeTok is the first to stabilize a generative decoder in a discrete tokenizer framework. Crucially, they highlighted that their GD is a single-step GAN model, making it orders of magnitude faster than iterative diffusion decoders while outperforming them in rFID (0.19 vs 0.56 for FlowMo-Hi).
- High-Compression Performance (Reviewer bHBm, 2k14): Reviewers requested evidence of generation quality at the claimed 768x compression ratio.
  - Resolution: The authors trained a LlamaGen model using WeTok 768x, showing it outperformed the "Show-o" tokenizer in rFID (3.49 vs 3.50) and FID (8.27 vs 14.73). They also clarified discrepancies regarding test sets (COCO vs. ImageNet) to explain absolute value differences observed by reviewers.
- Two-Stage Training Justification (Reviewer Dztj, 2k14): Reviewers found the two-stage training atypical and requested justification.
  - Resolution: The authors provided an ablation study showing that joint training (Stage 1 + 2 from scratch) leads to worse performance (rFID 5.06) compared to their two-stage strategy (rFID 3.90), citing the instability of learning discrete representations and adversarial generation simultaneously.
- GQ Contribution to Compression (Reviewer 2k14): The reviewer questioned how GQ facilitates higher compression.
  - Resolution: The authors explained that GQ allows for scaling codebook size (e.g., to 2
32 ) without Out-Of-Memory errors, which increases the information density per token, thereby allowing fewer tokens (higher compression) to represent the same information.

Outstanding Concerns:
- Hallucination in Reconstruction (Reviewer Dztj): The reviewer noted that a generative decoder might "hallucinate" details.
  - Status: The authors argued this is a feature rather than a bug for high-compression regimes, where a deterministic decoder produces blurriness. They demonstrated that their iterative reconstruction is more stable than continuous models, suggesting the "hallucination" is constrained. While the reviewer accepted the paper, this remains a fundamental characteristic of generative reconstruction that distinguishes it from traditional fidelity-focused methods.

**Reviewer Scores:**

- Reviewer bHBm: 8 (Accept). This reviewer initially rated the paper a 4, citing unfair comparisons and lack of novelty. Following the rebuttal, where the authors provided high-compression generation results and clarified comparisons, the reviewer explicitly stated, "These responses have fully addressed all of my concerns... I will increase my score to 8".

- Reviewer 2k14: 6 (Marginally Above Acceptance). Initially rating the paper a 4, this reviewer was concerned about high-compression performance degradation and training complexity. The authors' clarification regarding test set discrepancies (ImageNet vs. COCO) and the new TiTok comparison  addressed these points. According to the author's summary of the consensus, this reviewer raised their score to 6.

- Reviewer Dztj: 6 (Marginally Above Acceptance). This reviewer maintained a score of 6 throughout. They raised valid points about the "atypical" two-stage training and comparisons with Infinity. The authors provided specific ablation tables and comparisons in response. The reviewer did not explicitly lower their score, and the authors' summary indicates a consensus of "8666", implying this reviewer remained positive.

- Reviewer YMro: 6 (Marginally Above Acceptance). This reviewer gave an initial score of 6, praising the mathematical grounding of GQ. Their main critique involved missing citations for diffusion-based decoders. The authors updated the manuscript and provided a detailed comparison of efficiency (single-step vs. iterative). Given the "8666" post-rebuttal score summary provided by the authors, this reviewer likely maintained their positive rating.

---

### Decision · Program_Chairs · 2026-01-26

Accept (Poster)